# Changes to gut amino acid transporters and microbiome associated with increased E/I ratio in $Chd8^{+/-}$ mouse model of ASD-like behavior

You Yu[1,7], Bing Zhang[1,2,7], Peifeng Ji[1,7], Zhenqiang Zuo[1], Yongxi Huang[1], Ning Wang[1], Chang Liu [3], Shuang-Jiang Liu [3] & Fangqing Zhao [1,2,4,5,6✉]

Autism spectrum disorder (ASD), a group of neurodevelopmental disorders characterized by social communication deficits and stereotyped behaviors, may be associated with changes to the gut microbiota. However, how gut commensal bacteria modulate brain function in ASD remains unclear. Here, we used chromodomain helicase DNA-binding protein 8 (CHD8) haploinsufficient mice as a model of ASD to elucidate the pathways through which the host and gut microbiota interact with each other. We found that increased levels of amino acid transporters in the intestines of the mouse model of ASD contribute to the high level of serum glutamine and the increased excitation/inhibition (E/I) ratio in the brain. In addition, elevated α-defensin levels in the haploinsufficient mice resulted in dysregulation of the gut microbiota characterized by a reduced abundance of *Bacteroides*. Furthermore, supplementation with *Bacteroides uniformis* improved the ASD-like behaviors and restored the E/I ratio in the brain by decreasing intestinal amino acid transport and the serum glutamine levels. Our study demonstrates associations between changes in the gut microbiota and amino acid transporters, and ASD-like behavioral and electrophysiology phenotypes, in a mouse model.

[1] Beijing Institutes of Life Science, Chinese Academy of Sciences, Beijing, China. [2] University of Chinese Academy of Sciences, Beijing, China. [3] State Key Laboratory of Microbial Resources, Institute of Microbiology, Chinese Academy of Sciences, Beijing, China. [4] Key Laboratory of Systems Biology, Hangzhou Institute for Advanced Study, University of Chinese Academy of Sciences, Hangzhou, China. [5] State Key Laboratory of Integrated Management of Pest Insects and Rodents, Institute of Zoology, Chinese Academy of Sciences, Beijing, China. [6] Center for Excellence in Animal Evolution and Genetics, Chinese Academy of Sciences, Kunming, China. [7] These authors contributed equally: You Yu, Bing Zhang, Peifeng Ji. ✉email: zhfq@biols.ac.cn

Autism spectrum disorder (ASD), a group of heterogeneous neurodevelopmental disorders, is characterized by deficits in social communication and repetitive and stereotyped behaviors[1]. The reported incidence of ASD in children and adolescents is 1 in 59 in the United States[2,3]. ASD is believed to be influenced by genetic factors, including many de novo mutations and copy number variations identified by trio-based sequencing[4–6], and environmental factors, which may include the gut microbiota[7]. In recent years, accumulating studies have proven the effects of the gut microbiota on neural activities[8].

The gut microbiota has been shown to modulate the neural function and central nervous system (CNS)-related behaviors in physiological and pathological conditions[9]. In both people with ASD and animal models, gut microbiota dysbiosis has been reported[10,11], and probiotics may alleviate the symptoms[12]. Despite the importance of gut microbiota in modulating neural activities, how gut commensal bacteria interact with the brain in ASD is still unclear. *Lactobaciullus reuteri* was shown to modulate the oxytocinergic and dopaminergic signaling pathways through the vagus nerve in ASD mouse models[10]. In addition, the gut microbiota metabolite 5-aminovaleric acid could rescue ASD-like behaviors by decreasing the excitability of pyramidal neurons in layer V in BTBR mice[13]. However, the identity of pathways other than the vagus nerve axis and the neuroactive compounds that mediate the neural function of the gut microbiota are unknown.

Since *Chd8*, which encodes the chromatin remodeler CHD8, is one of the top genes associated with ASD and individuals with *Chd8* mutations exhibit gastrointestinal defects, *Chd8*$^{+/-}$ mice can be used as an ASD animal model to study the role of the gut microbiota in ASD and the underlying communication between the gut microbiota and the brain. Herein, we provided evidence for how the gut microbiota communicates with the brain via the neuroactive pathway in a model of ASD. Increased intestinal amino acid transport contributes to the increased excitation/inhibition (E/I) ratio with the increased glutamate levels in the brain, and the gut microbiota can restore the E/I imbalance in the brain by decreasing intestinal amino acid transport.

## Results

***Chd8*$^{+/-}$ mice exhibit anatomical abnormalities and ASD-like behaviors.** To generate a *Chd8* gene-deficient mouse model of autism, we deleted 13 bp from the first exon of the *Chd8* gene, resulting in a frameshift mutation and a premature stop codon in *Chd8* (Fig. 1a). Since the first exon is before most mutation sites in ASD linked to *Chd8* defects[14], the mouse model carrying a frameshift mutation at this position can mimic the effect of *Chd8* mutants in individuals with ASD. Given that *Chd8* homozygous mutant mice were reported to be embryonic lethal[15], we used heterozygous mice and genotyped them by specific PCR primers targeting the mutant allele (Supplementary Fig. 1a, b). Subsequently, to verify the validity of the model, we examined *Chd8* expression at the protein level; there are two isoforms, a full-length protein (280 kDa) and a smaller isoform (110 kDa). At the embryonic stage, in which *Chd8* expression is high, the *Chd8*$^{+/-}$ mice showed significantly reduced expression of the full-length CHD8 protein in the brains (the whole brain and cerebral cortex) and intestines (Fig. 1b and Supplementary Fig. 1c), while the 110 kDa isoform did not change significantly in the whole brain or the cerebral cortex (Supplementary Fig. 1c). At the adult stage, the expression of the full-length CHD8 protein was also reduced in both the hypothalamus (Fig. 1b) and the cerebral cortex (Supplementary Fig. 1d) but could not be detected in the small intestine. The adult hypothalamus and the cerebral cortex showed no significant change in the expression of the 110 kDa isoform (Supplementary Fig. 1d). The adult and embryonic small intestines showed no expression of the 110 kDa isoform.

Similar to patients with mutant *Chd8* exhibiting macrocephaly[14], the *Chd8*$^{+/-}$ mice showed increased brain weights, with the most significant increase observed in their cerebrum and cerebellum (Fig. 1c and Supplementary Fig. 1e). The *Chd8*$^{+/-}$ mice exhibited a shortened small intestine and colon length (Fig. 1d), a decreased weight of the small intestine contents (Supplementary Fig. 1e), and a trend towards decreased intestinal motility (Supplementary Fig. 1f). These observed abnormalities of the gastrointestinal tract were similar to those reported in humans, zebrafish, and mouse models with *Chd8* mutations[14,15]. The *Chd8*$^{+/-}$ mice also displayed weight loss at weeks 4, 12, and 18 (Fig. 1e), which was consistent with prior studies showing that individuals with ASD and mouse models with mutations in *Chd8* showed weight loss[14,16]. We evaluated the feeding behavior of *Chd8*$^{+/-}$ mice but found no significant changes in food intake (Supplementary Fig. 1g).

Next, we explored whether the *Chd8*$^{+/-}$ mice exhibited ASD-like behaviors, including impaired social interaction, anxiety, learning and memory deficits, stereotyped behavior, and depression. Social interaction was examined using a three-chamber test and a reciprocal social interaction test. The *Chd8*$^{+/-}$ mice exhibited normal sociability (Supplementary Fig. 1h) and abnormal social novelty preferences (Fig. 1f) in the three-chamber test and no significant social interaction deficits in the reciprocal social interaction test (Supplementary Fig. 1i), suggesting mildly impaired social interaction. Anxiety-like behavior was studied via an open-field test and a light/dark box test. The *Chd8*$^{+/-}$ mice showed a decreased proportion of time in the center in the open-field test (Fig. 1g and Supplementary Fig. 1j) and a significantly reduced duration in the light box in the light/dark box test (Fig. 1h), indicating obvious anxiety. Learning and memory were examined using a novel object recognition test. The *Chd8*$^{+/-}$ mice displayed no significant difference in the exploration time of the old and novel objects, demonstrating learning and memory deficits (Fig. 1i). Stereotyped behavior was explored through a self-grooming test and a marble-burying test. The *Chd8*$^{+/-}$ mice did not exhibit a significantly increased grooming time in the self-grooming test or a significant increase in buried marbles in the marble-burying test, indicating no significant difference in stereotyped behavior (Supplementary Fig. 1k). The *Chd8*$^{+/-}$ mice also showed no significant difference in total distance traveled or movement velocity in the open-field test (Supplementary Fig. 1l), indicating no significant abnormality in motor function. Depression was detected via the forced swimming test. The *Chd8*$^{+/-}$ mice displayed no significant increase in immobility time (Supplementary Fig. 1m). Female *Chd8*$^{+/-}$ mice showed ASD-like behaviors similar to those of the male *Chd8*$^{+/-}$ mice (Supplementary Fig. 1n–t). These results indicated that the *Chd8*$^{+/-}$ mice have ASD-like behaviors, including anxiety, impaired social interaction, and learning and memory deficits.

***Chd8*$^{+/-}$ mice exhibit an increased E/I ratio.** The *Chd8*$^{+/-}$ mice exhibited ASD-like behaviors, but the etiology of these behaviors remains unknown. Given that oxytocin was associated with ASD and that ASD mouse models showed decreased oxytocin levels[7,17], we investigated the oxytocin levels in the *Chd8*$^{+/-}$ mice. Immunofluorescence analysis did not show a significant difference in the level of oxytocin in the hypothalamic paraventricular nuclei (PVN) and the supraoptic nuclei (SON) (Supplementary Fig. 2a). In addition to the change in oxytocin, an E/I imbalance is one of the etiologies of ASD. An E/I imbalance has been reported in humans and several ASD animal models[18–20]. To evaluate the E/I imbalance in the *Chd8*$^{+/-}$ mice, we tested the miniature excitatory postsynaptic currents

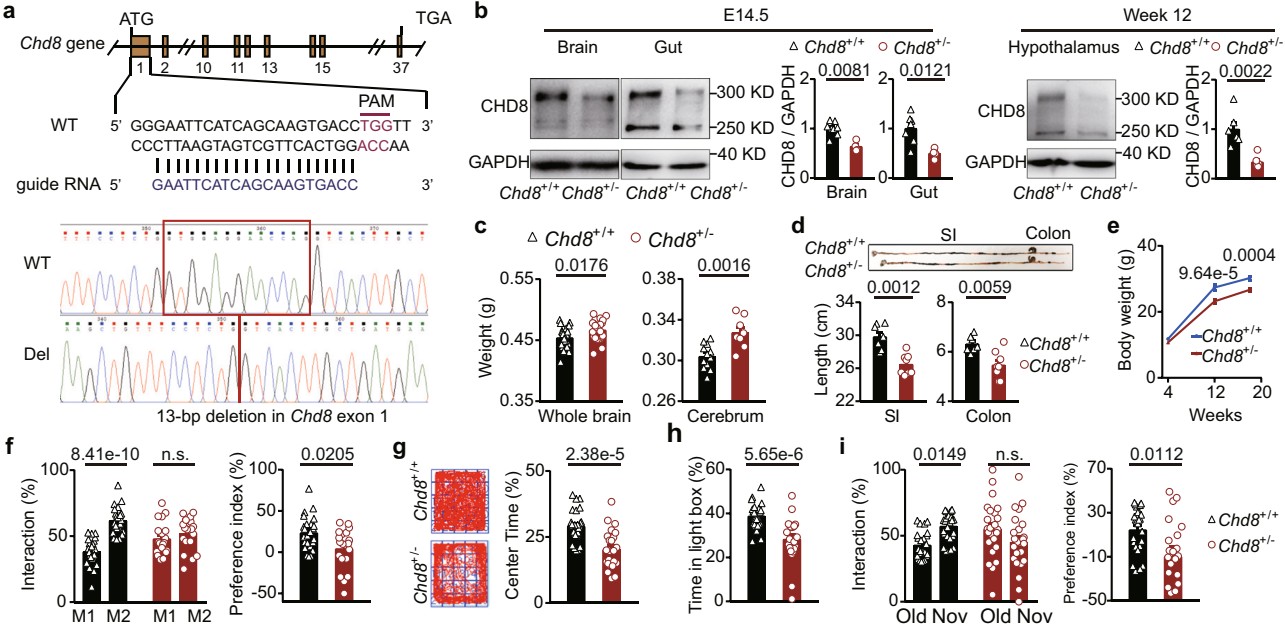

**Fig. 1 Chd8$^{+/-}$ mice show abnormalities in the brain and intestine and ASD-like behaviors. a** Chd8 mutant construction and the sequences of wild-type and mutant alleles. **b** Full-length CHD8 protein expression in the whole brain and the gut of the mice at embryonic day 14.5 (E14.5) as well as in the hypothalamus of the mice at week 12. $n = 8, 4, 7, 4, 6$ and 6 mice, respectively. **c** Weights of the whole brain and cerebrum of 12-week-old mice. $n = 26, 24$, 12, and 8 mice, respectively. **d** Lengths of the small intestine (SI) and colon of 12-week-old mice. $n = 7$ and 8 mice, respectively. **e** Body weights of the mice at weeks 4, 12, and 18. $n = 20$ (Week 4/Chd8$^{+/+}$), 23 (Week 4/Chd8$^{+/-}$), 12 (Week 12/Chd8$^{+/+}$), 11 (Week 12/Chd8$^{+/-}$), 14 (Week 18/Chd8$^{+/+}$), and 13 (Week 18/Chd8$^{+/-}$) mice, respectively. **f** Percentages of interaction time (left) and the preference index (right) in the social novelty preference session of the three-chamber social interaction test. $n = 24$ and 20 mice, respectively. **g** Representative traces (left) and the percentages of time in the center in the open-field test (right). $n = 26$ and 24 mice, respectively. **h** Percentages of time spent in the light box in the light/dark box test. $n = 26$ and 22 mice, respectively. **i** Percentages of interaction time (left) and the preference index (right) in the novel object recognition test. Nov novel object, Old old object. $n = 25$ and 24 mice, respectively. Source data are provided as a Source Data file. Quantitative data are shown as the mean ± SEM. Statistical analysis was determined by the two-tailed Mann–Whitney test (**b**, **c**, **d**, **f** (right panel), **g**, **h** and **i** (right panel)) and two-way ANOVA with two-tailed Turkey's test for multiple comparisons (**e**, **f** (left panel) and **i** (left panel)). Significance was indicated by P value. n.s. means no significant difference.

(mEPSCs) and the miniature inhibitory postsynaptic currents (mIPSCs) of pyramidal cells in neocortical layer V and observed an increased mEPSC frequency and a decreased mIPSC frequency but no significant differences in the amplitudes of the mEPSCs and mIPSCs (Fig. 2a, b). These results suggested that the Chd8$^{+/-}$ mice exhibit an E/I imbalance with an increased E/I ratio.

Next, we explored the mechanism underlying the E/I imbalance. Given that some amino acids can act as excitatory or inhibitory neurotransmitters or as neurotransmitter precursors and thus affect the E/I imbalance, we examined the levels of twenty-two common amino acids in the brain using a targeted metabolomics assay. The Chd8$^{+/-}$ mice showed increased levels of glutamate and glutamine and decreased levels of tryptophan and histidine in the brains (Supplementary Fig. 2b). Furthermore, the Chd8$^{+/-}$ mouse brains showed an increased glutamate/GABA ratio (Supplementary Fig. 2b), corroborating the finding of an increased E/I ratio at the neural circuit level (Fig. 2a, b). To explore these changes in different parts of the brain, we examined the glutamate, glutamine, and GABA levels in the cerebrum, brain stem, and cerebrum and found that the cerebrum displayed the most significant changes (Fig. 2c and Supplementary Fig. 2c).

Among the above amino acids, glutamate was selected, as it is the major excitatory neurotransmitter in the brain, and elevated glutamate levels indicate increased excitability[21]. We next explored the reasons for the increased glutamate level in the brain. Since glutamate cannot pass through the brain–blood barrier (BBB)[22], this amino acid is synthesized in situ in the brain by the conversion of other substances[23] (Supplementary Fig. 2d). Glutamate can be reversibly generated from glutamine, GABA,

and αKG. We found an elevated level of glutamine and no significant differences in the αKG (Supplementary Fig. 2e) and GABA levels (Supplementary Fig. 2b, c), suggesting that the increased glutamate level in the brain might result from an increase in glutamine.

We next explored the reasons for the increased glutamine level in the brain; this change may be due to the conversion of other substances in the brain as well as the transport of serum glutamine to the brain via the BBB. We first focused on factors outside the brain that can influence the glutamine level in the brain, including the expression levels of neutral amino acid transporters at the BBB and the serum glutamine concentration. We examined the gene expression of the large neutral amino acid transporter LAT1 (SLC7A5)[24,25] by single-cell RNA-seq (scRNA-seq) and found no significant difference in the expression level of Slc7a5 in endothelial cells of the Chd8$^{+/-}$ mice (Supplementary Fig. 2f, g). To test the serum glutamine level, we performed targeted metabolomics assays for twenty-two amino acids, including glutamine. The Chd8$^{+/-}$ mice showed significantly increased serum levels of glutamine and tryptophan (Fig. 2d and Supplementary Fig. 2h). These results suggest that increased serum glutamine in the Chd8$^{+/-}$ mice may contribute to the increased E/I ratio characterized by elevated glutamate levels and glutamate/GABA ratios.

**Increased expression of intestinal amino acid transporters may contribute to the elevated serum amino acid levels of the Chd8$^{+/-}$ mice.** We subsequently explored the mechanisms

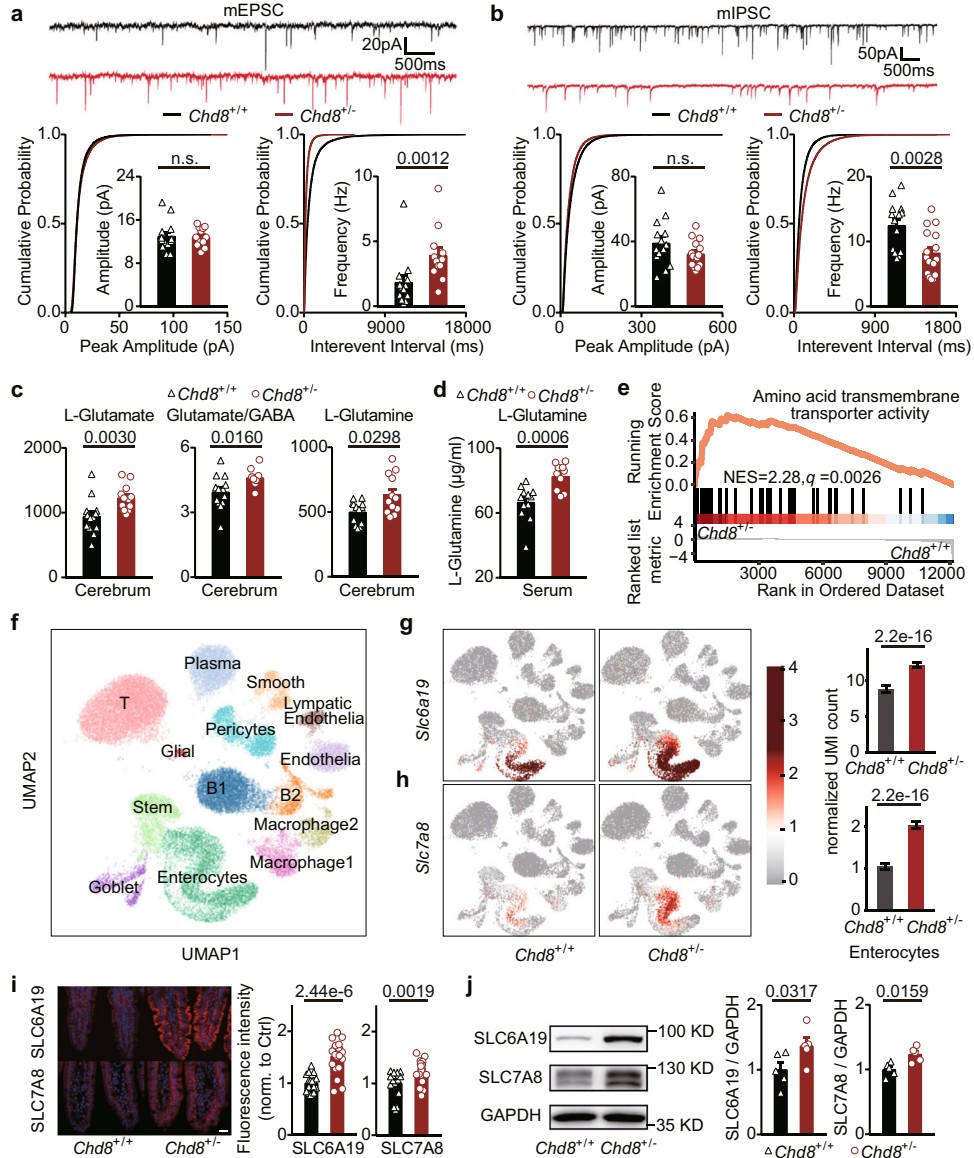

**Fig. 2 The E/I imbalance in the *Chd8*⁺/⁻ mice is associated with abnormalities in both the brain and intestine. a, b** Representative traces (top), cumulative distribution plots (bottom) and bar graphs (bottom) showing the amplitude and frequency of mEPSCs (**a**) and mIPSCs (**b**) of pyramidal cells in neocortical layer V of adult mice (weeks 8–9). $n = 3$ mice per group. Four to six data points per mouse. $n = 13$ (mEPSCs/*Chd8*⁺/⁺), 12 (mEPSCs/*Chd8*⁺/⁻), 14 (mIPSCs/*Chd8*⁺/⁺), and 15 (mIPSCs/*Chd8*⁺/⁻) data points, respectively. **c** Levels of L-glutamate (μg/g), glutamate/GABA, and L-glutamine (μg/g) in the cerebrum of 12-week-old mice detected by the targeted metabolomics assay. $n = 13$ and 12 mice, respectively. **d** Level of L-glutamine (μg/ml) in the serum of 12-week-old mice detected by the targeted metabolomics assay. $n = 12$ and 10 mice, respectively. **e** GSEA plot showing the enrichment of the amino acid transmembrane transporter activity pathway in small intestinal cells. **f** UMAP plot of single-cell transcriptome profiles of intestines from 12-week-old mice. **g, h** Highlights of enterocyte cells expressing *Slc6a19* (**g**) and *Slc7a8* (**h**) in the UMAP plot (left), and comparison of the expression levels of *Slc6a19* and *Slc7a8* between the *Chd8*⁺/⁺ and *Chd8*⁺/⁻ mice (right) ($n = 4$ mice). The bar plots represent the mean of abundance of *Slc6a19* (top) and *Slc7a8* (bottom) in enterocyte cells ($n = 1527$ and 3445 cells in *Chd8*⁺/⁺ and *Chd8*⁺/⁻ mice, respectively). Error bars represent 95% confidence interval (CI) of mean value. **i** Immunofluorescence staining of SLC6A19 and SLC7A8 in the ileum of the mice at week 12. Scale bar, 20 μm. Each symbol represents one image; six to seven images per mouse. $n = 17$ and 22 symbols, respectively. **j** Western blotting analysis of SLC6A19 and SLC7A8 protein levels in the small intestine of mice at week 12 ($n = 6$ mice). Source data are provided as a Source Data file. Quantitative data are shown as the mean ± SEM. Statistical analysis was determined by the two-tailed Mann–Whitney test (**a–d, g–j**). Significance was indicated by P value (**a–d, g–j**). n.s. means no significant difference.

underlying the elevated levels of serum glutamine and tryptophan in the *Chd8*⁺/⁻ mice. Intestinal transport of amino acids to the blood is one of the main sources of amino acids in the serum. Therefore, we examined the expression of amino acid transporters in the small intestines using bulk RNA-seq. The *Chd8*⁺/⁻ mice showed elevated expression of intestinal amino acid transporters (Fig. 2e and Supplementary Fig. 3a). Among these

transporters, the neutral amino acid transporters responsible for glutamine and tryptophan transport were examined (Supplementary Fig. 3b). We found elevated expression levels of *Slc6a19*, *Slc7a8*, and *Slc7a15* (Supplementary Fig. 3b) and validated them using quantitative PCR (qPCR) analyses (Supplementary Fig. 3c). We further verified the expression levels of amino acid transporters in intestinal epithelial cells using scRNA-seq experiments

and observed increased expression levels of the three selected intestinal amino acid transporters and increased numbers of cells expressing these transporters (Fig. 2f–h and Supplementary Fig. 3d, e). Subsequently, we selected *Slc6a19* and *Slc7a8*, two highly expressed transporters on the apical and basal lateral membranes of the intestinal epithelium, respectively, for protein detection. Both Western blot and immunofluorescence experiments confirmed the increased protein expression of the two transporters in the $Chd8^{+/-}$ mice (Fig. 2i, j). These results indicated that the increased expression of amino acid transporters in intestinal epithelial cells may lead to elevated serum amino acid levels in the $Chd8^{+/-}$ mice.

Since abnormal expression of intestinal amino acid transporters affects serum amino acid levels as well as fecal amino acid levels[26], we next examined the levels of amino acids in the feces. Interestingly, we found that the levels of five amino acids, including glutamine and tryptophan, were significantly reduced in the feces of the $Chd8^{+/-}$ mice (Supplementary Fig. 3f). These results suggest that high expression of intestinal amino acid transporters may lead to more amino acids being absorbed in the serum and fewer amino acids being retained in the feces.

**The gut microbiota is dysregulated in the $Chd8^{+/-}$ mice.** Given the reported role of the gut microbiota in ASD, we investigated the gut microbiota in the $Chd8^{+/-}$ mice. We performed metagenomic sequencing on the fecal DNA of adult mice to examine the gut microbiota. The beta diversity showed a substantial difference between the two groups of mice (Fig. 3a and Supplementary Fig. 4a). Furthermore, the $Chd8^{+/-}$ mice showed a significantly reduced abundance of *Bacteroides*, while the levels of other genera remained largely unchanged (Fig. 3b). In addition, at the species level, the abundances of *Bacteroides uniformis*, *Bacteroides acidifaciens* and *Bacteroides vulgatus* were significantly reduced in the $Chd8^{+/-}$ mice, with *B. uniformis* showing the largest fold change (Fig. 3c, d). Moreover, the alpha diversity and the microbiota composition at the phylum level did not differ significantly between the two groups of mice (Supplementary Fig. 4b, c).

To explore how changes in the gut microbiota affected the metabolism of intestinal contents, we performed a microbiota functional analysis. A total of 301 Kyoto Encyclopedia of Genes and Genomes (KEGG) orthologs (KOs) were found to be significantly enriched or depleted, most of which were involved in amino acid metabolism and metabolic signaling pathways (Supplementary Fig. 4d–f). Notably, eight KOs related to amino acids were significantly depleted in the $Chd8^{+/-}$ mice (Supplementary Fig. 4f), suggesting that the decreased amino acid levels in the feces of the $Chd8^{+/-}$ mice might be due to dysregulated gut microbiota.

We next investigated the presence of other metabolic changes in the intestines in addition to dysregulated gut microbiota. Short-chain fatty acids (SCFAs) have been associated with the gut–brain axis[27], but the two groups of mice showed no significant changes in the SCFA levels in their feces (Supplementary Fig. 4g). Intestinal barrier dysfunction and BBB damage were also reported as mechanisms of gut–brain communication[28,29], but the $Chd8^{+/-}$ mice did not exhibit increased intestinal permeability (Supplementary Fig. 4h) or BBB permeability (Supplementary Fig. 4i). These results indicated that the $Chd8^{+/-}$ mice showed a disturbance in the gut microbiota including a decreased abundance of *Bacteroides*.

**Increased level of alpha defensins contributes to gut microbiota dysbiosis.** We further explored how the host influenced the dysbiosis of gut microbiota in the $Chd8^{+/-}$ mice. Given that defensins, short peptides produced by intestinal Paneth cells, play a pivotal role in maintaining the homeostasis of the gut microbiota[30], we screened eight common defensins expressed in the small intestine. The $Chd8^{+/-}$ mice showed elevated expression of *Defa1* and *Defa2* (Fig. 3e) but unchanged levels of the other six defensins (Supplementary Fig. 4j). Since the protein levels of defensin α1 were higher than those of defensin α2 in the contents of the small intestine, cecum, and colon of the $Chd8^{+/+}$ mice (Fig. 3f), we focused on defensin α1. We found that the $Chd8^{+/-}$ mice exhibited increased protein levels of defensin α1 in the small intestinal and colonic contents (Fig. 3g).

To determine whether the increased defensin α1 level was sufficient to cause gut microbiota dysbiosis in the $Chd8^{+/-}$ mice, we treated the $Chd8^{+/+}$ mice by gavage with defensin α1 every day for 8 consecutive days and examined the gut microbiota in their feces (Fig. 3h). The defensin α1-gavaged mice showed an increased abundance of *Firmicutes* ($P = 0.027$) (Fig. 3i), and the beta diversity at the genus level showed a significant difference between the two groups of mice (Fig. 3j and Supplementary Fig. 4k). The $Chd8^{+/-}$ mice showed a decrease in the abundance of *Bacteroides* in the gut microbiota, but this decrease was not found in the defensin α1-treated mice (Supplementary Fig. 4l). The Bray-Curtis (BC) distance and the Euclidean distance between the defensin α1-treated mice and the $Chd8^{+/+}$ mice (D_W) was slightly greater than that between the defensin α1-treated mice and the $Chd8^{+/-}$ mice (D_K) (Fig. 3k and Supplementary Fig. 4m). In addition, the BC distance and the Euclidean distance between the water-treated mice and the $Chd8^{+/-}$ mice (H_K) were greater than that between the water-treated mice and the $Chd8^{+/+}$ mice (H_W) (Supplementary Fig. 4n). These results indicate that the shifts in gut microbiota caused by defensin α1 treatment are partially towards ASD-like microbiota. To detect whether the influence of defensins on the microbiota, especially the level of *B. uniformis*, was a cumulative effect of multiple defensins, we treated the $Chd8^{+/+}$ mice with defensin α1 and α2 every day by gavage for 8 consecutive days (Fig. 3l). We found that the defensin α1 and α2-treated mice showed a slightly decreased level of *B. uniformis* in their feces (Fig. 3l).

To further investigate the role of alpha defensins in the gut microbiota, especially the level of *B. uniformis*, we decreased the level of alpha defensins in the intestine of $Chd8^{+/-}$ mice by using an inhibitor of matrix metalloproteinase (MMP), GM6001 (Fig. 3m), since alpha defensin precursors are cleaved and activated by MMP7[31,32]. We found that $Chd8^{+/-}$ mice intraperitoneally injected with GM6001 showed increased intestinal alpha defensin levels and decreased *B. uniformis* abundance (Fig. 3n and Supplementary Fig. 4o, p). Taken together, these results indicated that increased levels of alpha defensins contribute to gut microbiota dysbiosis, especially the level of *B. uniformis*, in the $Chd8^{+/-}$ mice.

In addition to defensins, intestinal inflammation can cause gut microbiota dysbiosis[33,34]. We further examined eight intestinal inflammatory cytokines related to the microbiota and found that the levels of six proinflammatory cytokines increased (Fig. 3o). The $Chd8^{+/-}$ mice showed increased levels of intestinal inflammatory cytokines but unchanged serum inflammatory cytokines (Supplementary Fig. 4q), suggesting that the altered microbial composition in $Chd8^{+/-}$ mice may be related to intestinal inflammation. Since dysbiosis in the gut microbiota can also lead to inflammation[33], gut microbiota dysbiosis and increased inflammation in $Chd8^{+/-}$ mice may reinforce each other.

**$B.\ uniformis$ improves the ASD-like behaviors and restores E/I imbalance in the $Chd8^{+/-}$ mice.** To reveal whether the

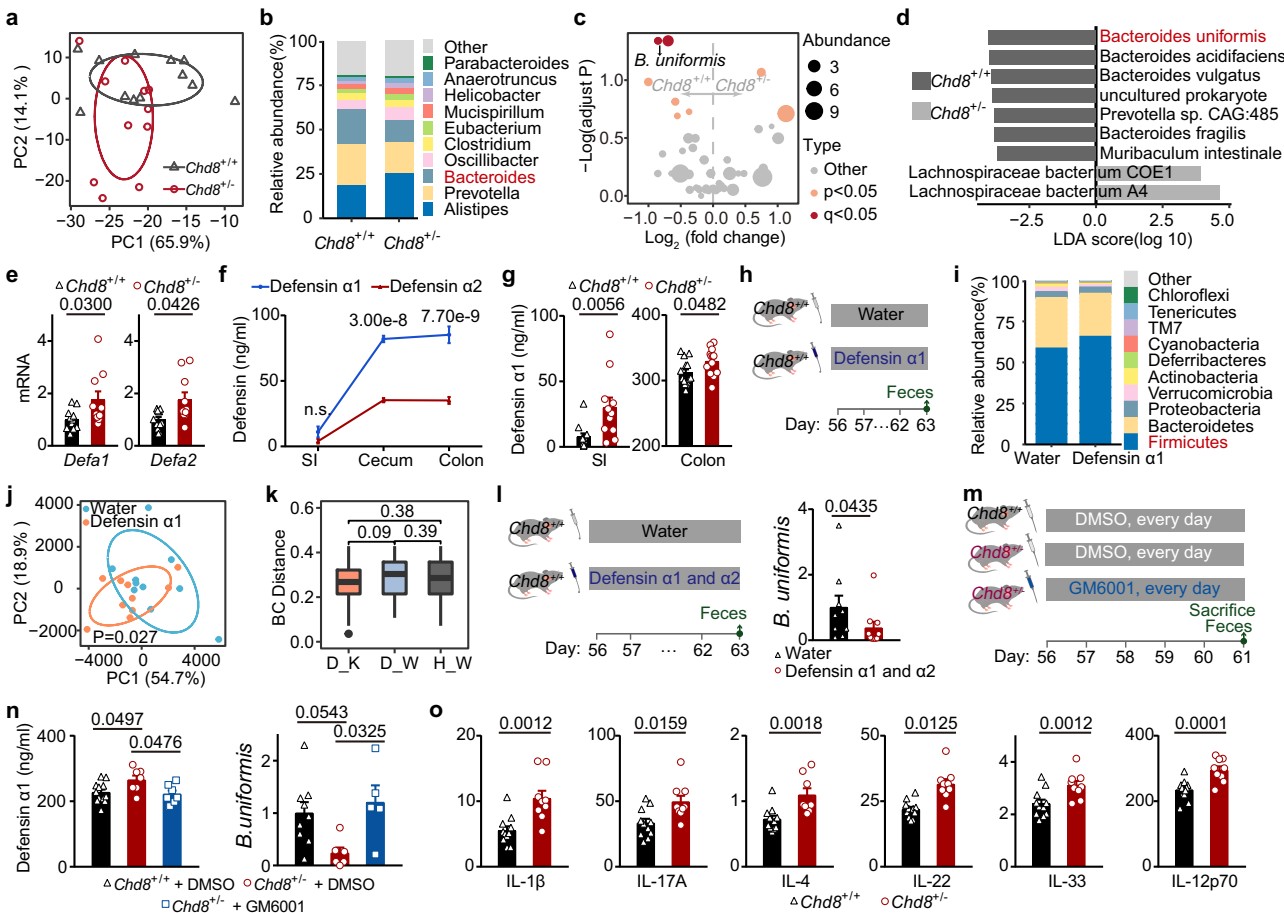

**Fig. 3 Increased levels of alpha defensins contribute to gut microbiota dysbiosis in the $Chd8^{+/-}$ mice. a–d** Metagenomic sequencing of the gut microbiota from 8-week-old mice. $n = 12$ and 11 mice, respectively. **a** PCA plots (species, Total Sum Scaling-Transformed). **b** Microbiota composition (genus). **c** Volcano plots. **d** LDA plots. **e** qPCR analysis in the small intestine of 12-week-old mice. $n = 12$ and 10 mice, respectively. **f** ELISA of proteins in the small intestinal, cecal, and colonic contents of 12-week-old $Chd8^{+/+}$ mice ($n = 5$). **g** ELISA of defensin α1 in the small intestinal (SI) and colonic contents (Colon) of 12-week-old mice. $n = 12$, 11, 14, and 13 mice, respectively. **h** Schematic diagram for the defensin α1 gavage. **i–k** A 16S rRNA sequencing of gut microbiota from 63-day-old mice ($n = 10$ mice). **i** Microbiota composition (phylum). **j** PCA plots (genus, Total Sum Scaling-Transformed) (ANOSIM test). **k** BC distance (Total Sum Scaling-Transformed) between D_W (defensin α1-treated mice and $Chd8^{+/+}$ mice), H_W (water-treated mice and $Chd8^{+/+}$ mice), and D_K (defensin α1-treated mice and $Chd8^{+/-}$ mice). Box plots were based on 350 data points and showed center line as median, box limits as upper and lower quartiles, whiskers as 1.5 × interquartile range and dots as outliers. **l** Schematic diagram for the defensin α1 and α2 gavage, and qPCR analysis of *B. uniformis* in the feces. $n = 9$ and 10 mice, respectively. **m** Schematic diagram for the GM6001 administration. **n** ELISA of defensin α1 in the cecal contents (left), and qPCR analysis of *B. uniformis* in the feces (right). $n = 9$, 6, 5, 12, 7, and 7 mice, respectively. **o** Multiplexed determination of cytokine levels from colon of 12-week-old mice (pg/mg protein). $n = 11$ and 10 mice, respectively. Source data are provided as a Source Data file. Ellipses in (**a** and **j**) were drawn around each group's centroid (95%). Quantitative data are shown as the mean ± SEM. Statistical analysis was determined by the two-tailed Mann–Whitney test (**e**, **g**, **l**, and **o**), one-way ANOVA with two-tailed Tukey's multiple comparison test (**n**), and two-way ANOVA with Turkey's test for multiple comparisons (**f**). Significance was indicated by P value.

dysregulated gut microbiota would affect the behaviors of $Chd8^{+/-}$ mice, we co-housed the $Chd8^{+/+}$ and $Chd8^{+/-}$ mice after weaning in a 3:1 ratio (Fig. 4a), since it has been reported that housing conditions may impact both the microbiota and behaviors of animals[7]. After co-housing the mice for 4 weeks, we checked the gut microbiome and the behaviors of $Chd8^{+/+}$ mice, $Chd8^{+/-}$ mice, and co-housed $Chd8^{+/-}$ mice (Fig. 4a). We used the abundance change of *B. uniformis* as an indicator of the microbiota changes (Supplementary Fig. 5a), since it showed the largest change fold and a significantly reduced level in $Chd8^{+/-}$ mice (Fig. 3c, d). We found that co-housed $Chd8^{+/-}$ mice showed an increased level of *B. uniformis* in their feces compared with $Chd8^{+/-}$ mice (Fig. 4b). For behavioral tests, co-housing improved the anxiety-like behavior (Fig. 4c) and social interaction behavior (Fig. 4d), but failed to improve the learning and

memory deficits of $Chd8^{+/-}$ mice (Supplementary Fig. 5b). These results indicate the pivotal role of the gut microbiota in the ASD-like phenotypes of the $Chd8^{+/-}$ mice.

Next, to clarify the role of *B. uniformis* in the ASD-like phenotypes of the $Chd8^{+/-}$ mice, we treated weaned $Chd8^{+/-}$ mice by gavage with *B. uniformis* every 3 days for 4 weeks and then examined the effects of this treatment (Fig. 4e). Following gavage, the $Chd8^{+/-}$ mice showed an increased *B. uniformis* level in their feces (Supplementary Fig. 5c) but their anatomical abnormalities, including those in their body weight, brain weight, and cerebrum weight, were not changed (Supplementary Fig. 5d). Subsequently, we conducted a three-chamber test to detect the effects of recovery of *B. uniformis* on the social behavior of the $Chd8^{+/-}$ mice and found that the impaired social interaction, one of the core phenotypes of ASD models,

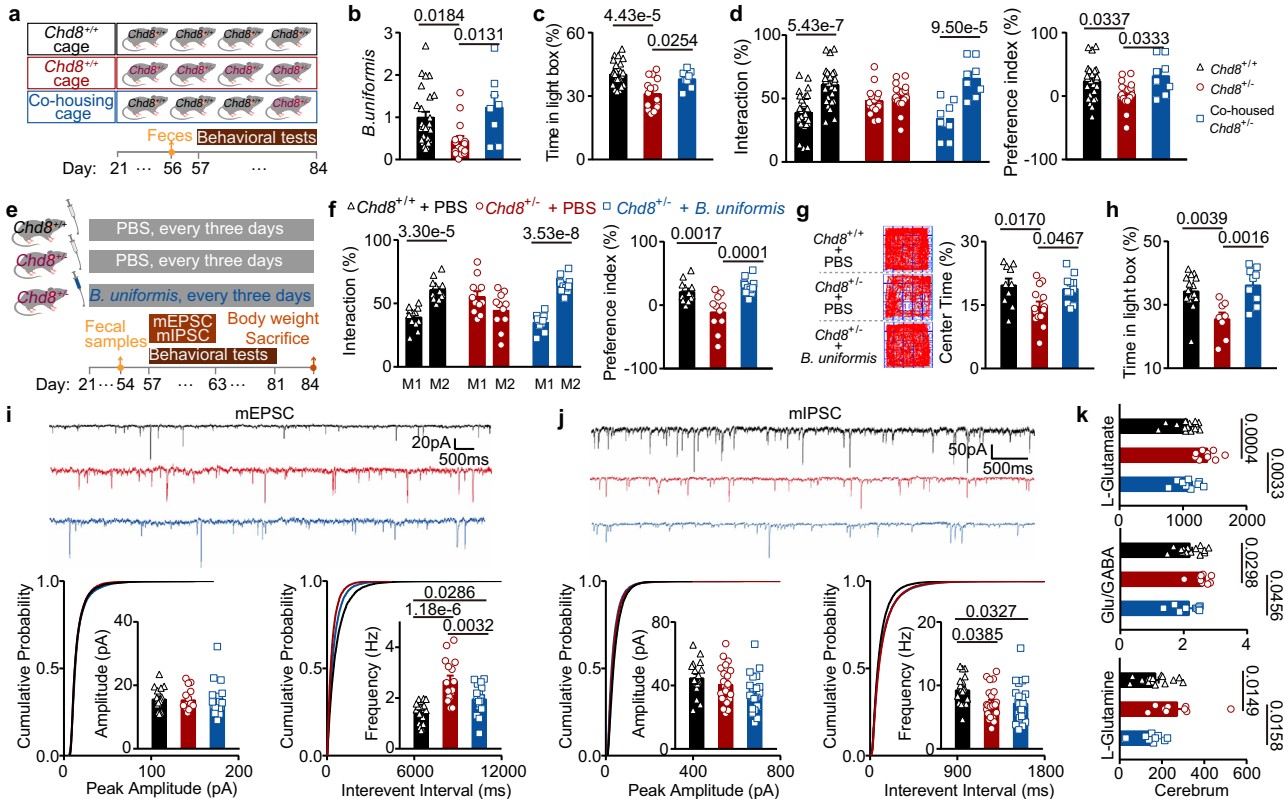

**Fig. 4 *B. uniformis* improves the ASD-like behaviors and restores E/I imbalance in the *Chd8*$^{+/-}$ mice. a** Schematic diagram for co-housing experiments. **b** qPCR analysis of *B. uniformis* in the feces from 8-week-old mice. $n = 26$, 16, and 8 mice, respectively. **c** Percentages of time spent in the light box in the light/dark box test. $n = 27$, 15, and 8 mice, respectively. **d** Percentages of the interaction time (left) and the preference index (right) in the social novelty preference session of the three-chamber social interaction test. $n = 26$, 18, and 8 mice, respectively. **e** Schematic diagram for *B. uniformis* gavage. **f** Percentages of the interaction time (left) and the preference index (right) in the social novelty preference session of the three-chamber social interaction test. $n = 12$, 11, and 11 mice, respectively. **g** Representative traces (left) and percentages of time in the center (right) in the open-field test. $n = 10$, 14, and 11 mice, respectively. **h** Percentages of time spent in the light box in the light/dark box test. $n = 15$, 8, and 9 mice, respectively. **i, j** Representative traces (top), cumulative distribution plots (bottom) and bar graphs (bottom) showing the amplitude and frequency of the mEPSCs (**e**) and the mIPSCs (**f**) of pyramidal cells in neocortical layer V. $n = 4$, 4, and 5 mice, respectively. Four to six data points per mouse. For mEPSCs, $n = 17$, 16, and 20 data points, respectively; for mIPSCs, $n = 18$, 22, and 25 data points, respectively. **k** Levels of L-glutamate (μg/g), glutamate/GABA (Glu/GABA), and L-glutamine (μg/g) in the cerebrum detected by targeted metabolomics assays. $n = 16$, 9, and 9 mice, respectively. Quantitative data are shown as the mean ± SEM. Statistical analysis was determined by one-way ANOVA with two-tailed Tukey's multiple comparison test (**b**, **c**, **d** (right panel), **f** (right panel), **g**, **h**, **i**, **j**, **k**) and two-way ANOVA with two-tailed Turkey's test for multiple comparisons (**d** (left panel) and **f** (left panel)). Significance was indicated by *P* value.

was improved (Fig. 4f). We also performed open-field and light/dark box tests to detect the effects of recovery of *B. uniformis* on the anxiety-like behavior of the *Chd8*$^{+/-}$ mice. Following gavage with *B. uniformis*, the *Chd8*$^{+/-}$ mice showed an increased proportion of time in the center in the open-field test and a significantly increased time in the light box in the light/dark box test (Fig. 4g, h), indicating the decreased anxiety-like behavior. The novel object recognition test revealed that *B. uniformis* failed to improve the learning and memory deficits in the *Chd8*$^{+/-}$ mice (Supplementary Fig. 5e). In addition, *B. uniformis* treatment did not impact the locomotor behaviors of *Chd8*$^{+/-}$ mice (Supplementary Fig. 5f).

In addition to behavioral phenotypes, the E/I imbalance of the *Chd8*$^{+/-}$ mice was also recovered by reducing the frequency of the mEPSCs (Fig. 4i, j). We further examined the effect of *B. uniformis* on the glutamate, GABA, and glutamine levels in the brains of the *Chd8*$^{+/-}$ mice since changes in glutamate levels and the glutamate/GABA ratio were one of the causes and manifestations of the E/I imbalance. *B. uniformis* reduced the glutamate and glutamine levels and the glutamate/GABA ratio in the cerebrum and the whole brain of the *Chd8*$^{+/-}$ mice (Fig. 4k and Supplementary Fig. 5g–i). Taken together, these results

indicated that *B. uniformis* can improve the ASD-like behaviors and restore E/I imbalance in the *Chd8*$^{+/-}$ mice.

**B. uniformis restores intestinal amino acid transporter levels.** The elevated serum glutamine level is believed to be one of the reasons for the increased glutamine level in the brain. To examine how *B. uniformis* affected the serum levels of amino acids, especially glutamine and tryptophan, we performed a targeted metabolomics assay for amino acids. We found that *B. uniformis* reduced the serum glutamine and tryptophan levels in the *Chd8*$^{+/-}$ mice without affecting the other amino acids (Fig. 5a and Supplementary Fig. 6a). To explore the effect of *B. uniformis* on the intestinal amino acid transporters of the *Chd8*$^{+/-}$ mice, we performed RNA-seq analysis of small intestinal cells. For the dysregulated intestinal genes, supplementation with *B. uniformis* decreased the differences between the *Chd8*$^{+/-}$ and *Chd8*$^{+/+}$ mice (Fig. 5b, c). In terms of enriched pathways of differentially expressed (DE) genes, supplementation with *B. uniformis* reduced the expression of intestinal amino acid transporters in the *Chd8*$^{+/-}$ mice but not to the same level as that in the *Chd8*$^{+/+}$ mice (Fig. 5d and Supplementary Fig. 6b–d). In terms of individual gene expression, the expression

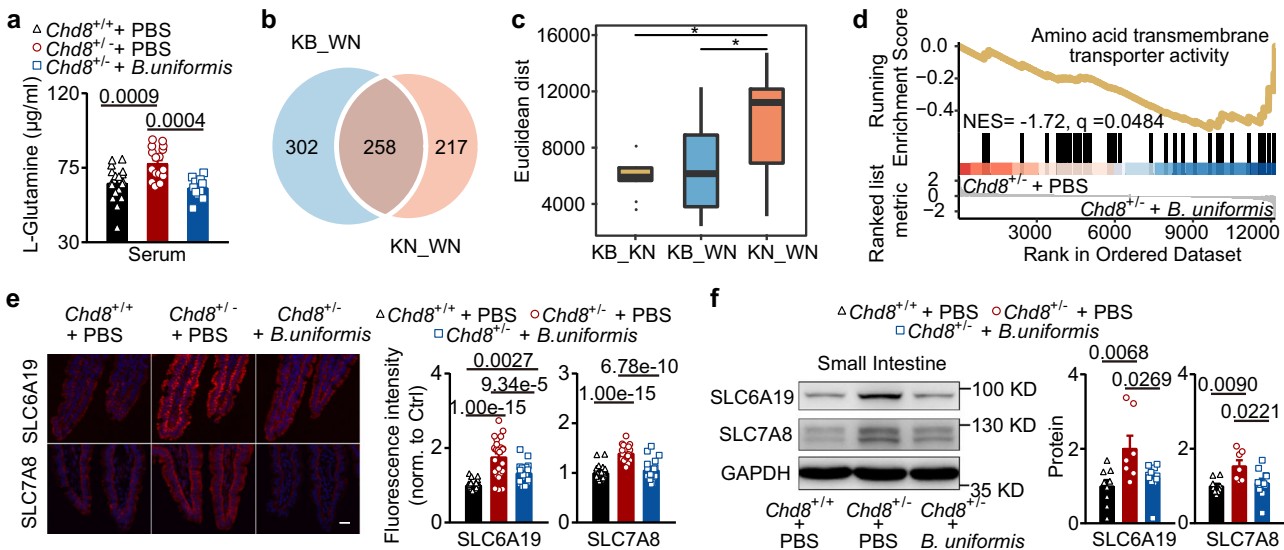

**Fig. 5** *B. uniformis* reduces the expression of intestinal amino acid transporters in the *Chd8*$^{+/-}$ mice. **a** Serum levels of L-glutamine in of 12-week-old mice detected by targeted metabolomics assays. $n = 18$, 18, and 11 mice, respectively. **b–d** Bulk RNA-seq analysis of the small intestine of 12-week-old mice. There were $n = 3$, $n = 3$, and $n = 5$ mice in the KN, KB, and WN groups, respectively (WN: *Chd8*$^{+/+}$ mice gavaged with PBS; KN: *Chd8*$^{+/-}$ mice gavaged with PBS; KB: *Chd8*$^{+/-}$ mice gavaged with *B. uniformis*). **b** Venn diagram illustrating the count of DE genes between KN_WN and DE genes between KB_WN. The DE genes were determined under a strict threshold of adjusted $P < 0.01$ and |log2(cf)| > 0.585. **c** Euclidean distance of KB_KN (*Chd8*$^{+/-}$ mice gavaged with PBS and *Chd8*$^{+/-}$ mice gavaged with *B. uniformis*), KB_WN, and KN_WN. The two-tailed Mann–Whitney test was used to calculate the significance. Box plots were based on 350 data points and showed center line as median, box limits as upper and lower quartiles, whiskers as 1.5 × interquartile range and dots as outliers. **d** GSEA plot shows the enrichment of the amino acid transmembrane transporter activity pathway in small intestinal cells of the *Chd8*$^{+/-}$ mice gavaged with PBS compared to the *Chd8*$^{+/-}$ mice gavaged with *B. uniformis*. **e** Immunofluorescence staining of SLC6A19 and SLC7A8 in the ileum of the mice at week 12. Scale bar, 20 μm. $n = 3$ mice per group. Each symbol represents one image; seven to nine images per mouse. $n = 28$, 26, and 21 symbols, respectively. **f** Western blotting analysis of the SLC6A19 and SLC7A8 protein levels in the small intestine of the mice at week 12. $n = 9$, 7, and 11 mice, respectively. Quantitative data are shown as the mean ± SEM. Statistical analysis was determined by one-way ANOVA with two-tailed Tukey's multiple comparison test (**a**, **e**, **f**). Significance was indicated by $P$ value.

levels of *Slc6a19*, *Slc7a8*, and *Slc7a15*, three highly expressed neutral amino acid transporters in the *Chd8*$^{+/-}$ mice, were reduced (Supplementary Fig. 6e), and these results were confirmed by qPCR assays (Supplementary Fig. 6f). At the protein level, SLC6A19 and SLC7A8 expression was also reduced by *B. uniformis* (Fig. 5e, f). Furthermore, supplementation with *B. uniformis* affected the fecal amino acid levels, in which the reduced fecal levels of several amino acids in the *Chd8*$^{+/-}$ mice were partially restored by supplementation with *B. uniformis* (Supplementary Fig. 6g). These results revealed that *B. uniformis* reduces the serum levels of amino acids by inhibiting intestinal amino acid transporter expression in the *Chd8*$^{+/-}$ mice.

We next explored whether the increased levels of amino acid transporters were due to the genetic mutation or the changes of the microbiome in *Chd8*$^{+/-}$ mice. We transferred the feces from *Chd8*$^{+/+}$ and *Chd8*$^{+/-}$ mice separately to the germ-free (GF) mice (Supplementary Fig. 6h) and then measured the level of amino acid transporters in the small intestines after fecal microbiota transplantation. We observed a slight decrease of *B. uniformis* in mice transferred with the feces of *Chd8*$^{+/-}$ mice (Supplementary Fig. 6i), but no significant alterations of amino acid transporters *slc6a19* and *slc7a8* in mRNA (Supplementary Fig. 6j) or protein level (Supplementary Fig. 6k), suggesting that dysbiotic gut microbiota alone cannot increase the expression of intestinal amino acid transporters without genetic mutation.

**An inhibitor of intestinal amino acid transporters restores ASD-like phenotypes**. We next explored whether *B. uniformis*-mediated modulation of intestinal amino acid transport could restore the ASD-like phenotypes in the *Chd8*$^{+/-}$ mice. First, we aimed to increase intestinal amino acid transport in the *Chd8*$^{+/+}$

mice and investigate whether these mice exhibited ASD-like phenotypes featuring an increased E/I ratio. The *Chd8*$^{+/+}$ mice were supplemented with glutamine for 2 weeks to increase the intestinal supply of glutamine (Supplementary Fig. 7a), but the serum glutamine level was not increased, even supplemented with the high concentration of glutamine (Supplementary Fig. 7b). Considering the influence of intestinal amino acid transporters on the serum levels of amino acids, we detected the expression of intestinal amino acid transporters. We found that a high concentration of glutamine in the drinking water reduced the expression of amino acid transporters to some extent (Supplementary Fig. 7c, d). Next, we intended to reduce intestinal amino acid transport in the *Chd8*$^{+/-}$ mice and investigated whether ASD-like phenotypes characterized by an increased E/I ratio were alleviated. In addition to the expression level, the activity of intestinal amino acid transporters affects amino acid transport. S-benzyl-L-cysteine[35] and nimesulide[36] have been previously reported to inhibit the intestinal amino acid transporter SLC6A19 at the cellular level (Fig. 6a). We added these two drugs to the drinking water of the *Chd8*$^{+/+}$ mice and tested the drugs' inhibitory effect 2 weeks later, using the serum glutamine level as an indicator (Fig. 6b). We found that both drugs could reduce the serum glutamine level and that the inhibitory effect of S-benzyl-L-cysteine was stronger than that of nimesulide (Fig. 6c). Subsequently, we investigated the effect of S-benzyl-L-cysteine on the glutamine and glutamate levels in the brain. We found that S-benzyl-L-cysteine reduced the level of glutamine, but not glutamate, in the brain of the *Chd8*$^{+/+}$ mice (Fig. 6d).

To explore the effect on the *Chd8*$^{+/-}$ mice, we added S-benzyl-L-cysteine in the drinking water of weaning *Chd8*$^{+/-}$ mice for 4 weeks and performed behavioral tests at the age of

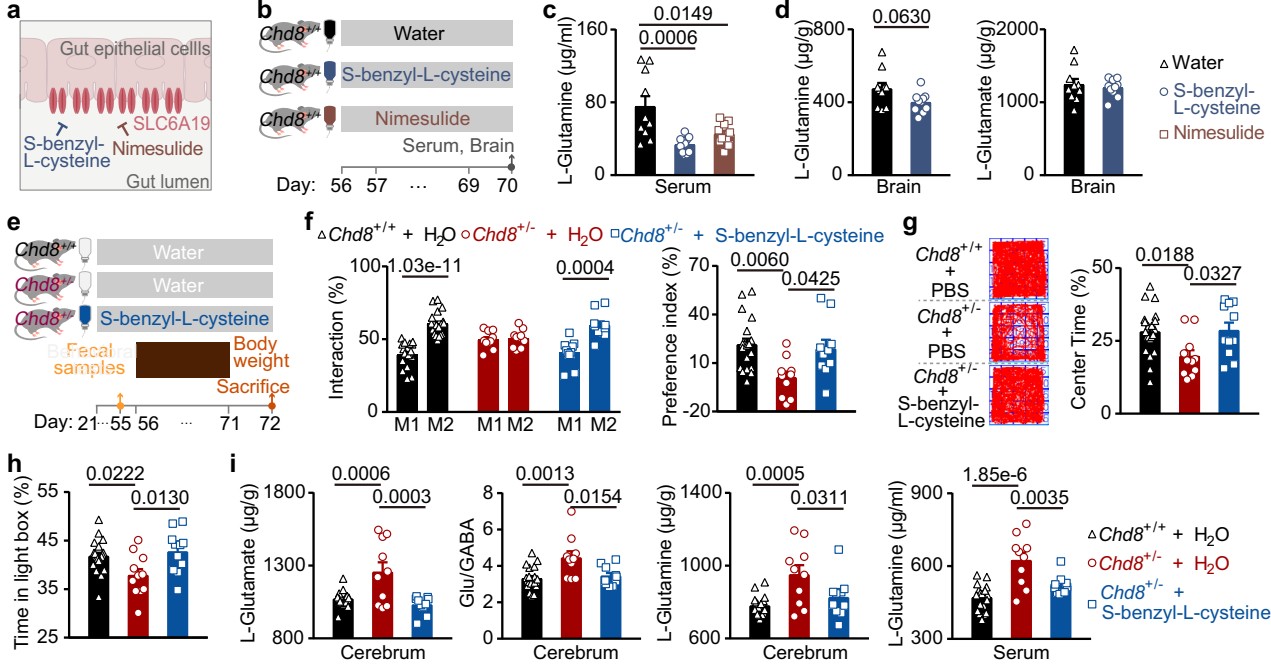

**Fig. 6 S-Benzyl-L-cysteine improves the ASD-like behaviors and restores E/I imbalance in the *Chd8*$^{+/-}$ mice. a** Schematic diagram showing S-benzyl-L-cysteine and nimesulide as inhibitors of the intestinal amino acid transporter SLC6A19. **b** Schematic diagram for supplementation of S-benzyl-L-cysteine and nimesulide in the drinking water of 8-week-old *Chd8*$^{+/+}$ mice for 14 days. $n = 10$, 11, and 10 mice, respectively. **c** Serum levels of L-glutamine detected by the targeted metabolomics assays ($n = 10$ mice). **d** Levels of L-glutamine and L-glutamate in the whole brain detected by the targeted metabolomics assays ($n = 10$ mice). **e** Schematic diagram for supplementation of S-benzyl-L-cysteine in the drinking water of 21-day-old *Chd8*$^{+/+}$ and *Chd8*$^{+/-}$ mice. **f** Percentages of the interaction time (left) and the preference index (right) in the social novelty preference session of the three-chamber social interaction test. $n = 19$, 10, and 10 mice, respectively. **g** Percentages of time spent in the light box in the light/dark box test. $n = 21$, 11, and 11 mice, respectively. **h** Representative traces (left) and percentages of time in the center (right) in the open-field test. $n = 21$, 11, and 11 mice, respectively. **i** Levels of L-glutamate, glutamate/GABA (Glu/GABA) and L-glutamine in the cerebrum and the level of L-glutamine in the serum detected by targeted metabolomics assays. $n = 20$, 10, and 10 mice, respectively. Source data are provided as a Source Data file. Quantitative data are shown as the mean ± SEM. Statistical analysis was determined by the two-tailed Mann–Whitney test (**d**), one-way ANOVA with two-tailed Tukey's multiple comparison test (**c**, **f** (right panel), **g**, **h**, **i**), and two-way ANOVA with two-tailed Turkey's test for multiple comparisons (**f** (left panel)). Significance was indicated by $P$ value.

8 weeks (Fig. 6e). S-benzyl-L-cysteine failed to recover the learning and memory deficits of the *Chd8*$^{+/-}$ mice (Supplementary Fig. 7e) but improved their impaired social interaction and anxiety (Fig. 6f–h), similar to the effect of *B. uniformis*. Furthermore, we found that S-benzyl-L-cysteine, similar to *B. uniformis*, could reduce the serum glutamine levels as well as the glutamine and glutamate levels and the glutamate/GABA ratio in the cerebrum of the *Chd8*$^{+/-}$ mice (Fig. 6i). S-benzyl-L-cysteine did not significantly affect the *Chd8*$^{+/-}$ mouse body weight, whole brain weight or cerebrum weight (Supplementary Fig. 7f). These results indicated that the restoration of the ASD-like phenotypes in the *Chd8*$^{+/-}$ mice by *B. uniformis* may be achieved by suppressing the expression of their intestinal amino acid transporters.

## Discussion

The microbiota–gut–brain axis, a bidirectional communication pathway linking the gut and brain, plays an important role in neuropsychiatric disorders. Nonetheless, the role of the microbiota–gut–brain axis in ASD is still unclear. Our study focused on analyzing the mutual interactions between the host and the gut microbiota in the development of ASD and our results support a role of intestinal amino acid transporters in the regulation of host neural activities by the gut microbiota in a mouse model of ASD-like behavior (Fig. 7).

We observed gut microbiota dysbiosis characterized by a reduced abundance of *Bacteroides* in the *Chd8*$^{+/-}$ mice, in consistent with previous studies that the gut microbiota in ASD exhibits a significant decrease in a large variety of probiotics, such as *Lactobacillus* and *Bacteroides*[7,37,38]. The supplementation of *Bacteroides fragilis* to the offspring of maternal immune activation mouses has shown beneficial effects on the CNS-related behaviors of the host[38]. Our study found that several species in *Bacteroides* exhibited reduced abundance in the *Chd8*$^{+/-}$ mice, and *B. uniformis* can rescue the ASD-relevant phenotypes and E/I imbalance. Furthermore, we found a slightly increased abundance of *Lachnospiraceae bacterium A4* in the *Chd8*$^{+/-}$ mice. *Lachnospiraceae bacterium A4* has reportedly inhibited Th2-cell differentiation in lamina propria, which is important in the host mucosal T-cell responses[39,40]. How *Lachnospiraceae bacterium A4* influences the host in the *Chd8*$^{+/-}$ mice and whether it affects the host neural activities by modulating mucosal immunity still need further investigation.

The host can regulate the gut microbiota through several direct or indirect pathways, among which antimicrobial peptides are one of the most direct ones[41]. We identified that an increased level of α-defensins contributed to gut microbiota dysbiosis in the *Chd8*$^{+/-}$ mice, in agreement with previous research demonstrating that α-defensins play a pivotal role in governing the intestinal microbial ecology[42]. Furthermore, defensins and CHD8 are expressed at different developmental stages in the intestine. Therefore, the regulation of defensins by CHD8 may be through indirect pathways. Extensive studies have shown that the Wnt-signaling pathway transcription factor Tcf-4 is required to express most α-defensins, and this process is independent of the gut microbiota[43–45]. Moreover, CHD8 is required for maintaining

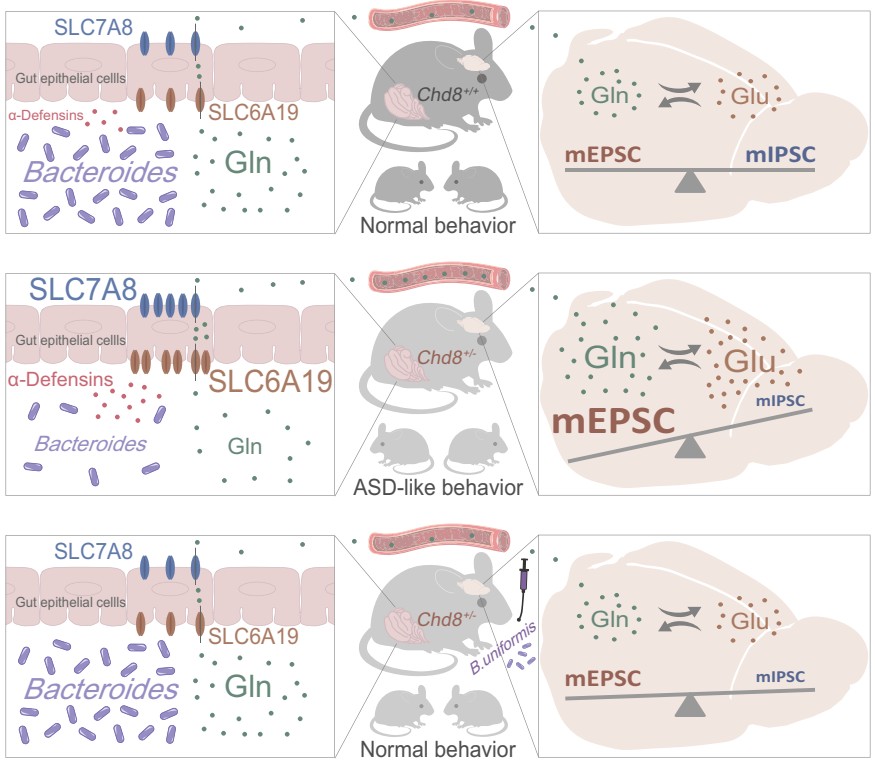

**Fig. 7 Summary of the microbiota–gut–brain axis in the CHD8 mouse model of ASD.** $Chd8^{+/-}$ mice show increased expression of the intestinal transporters SLC6A19 and SLC7A8, resulting in more glutamine transported to the serum. Elevated serum glutamine increases the levels of glutamine and glutamate in the brain, contributing to the E/I imbalance and ASD-like behaviors in the $Chd8^{+/-}$ mice. Gut microbiota dysbiosis characterized by a low abundance of *Bacteroides* in the $Chd8^{+/-}$ mice is induced by high levels of intestinal α-defensins. In addition, supplementation with *B. uniformis* improves the E/I imbalance and restores ASD-like behaviors in the $Chd8^{+/-}$ mice by decreasing the intestinal amino acid transport of glutamine and thus reducing the glutamine and glutamate levels in the brain.

Wnt signaling in mouse cortical neurons[46]. Whether CHD8 modulates elevated levels of defensins by influencing the Wnt/Tcf-4 remains to be clarified. In addition, we observed a slightly decreased intestinal transit in $Chd8^{+/-}$ mice. Since the change in gastrointestinal transit is supposed to affect the delivery rate of nutrients to the gut microbiota[41], the host may regulate the gut microbiota indirectly by changing the luminal environment in $Chd8^{+/-}$ mice.

We observed that the $Chd8^{+/-}$ mice show an E/I imbalance characterized by an increased glutamate/GABA ratio caused by elevated glutamate and glutamine levels in the brain. Except for the elevated glutamate and glutamine levels, gamma-glutamylated amino acid levels in the brain, which the gut microbiota can also modulate, reportedly have been associated with glutamate/GABA ratio[47]. Whether *B. uniformis* affects the glutamate/GABA ratio by regulating gamma-glutamylated amino acids in the brain needs further studies. In this study, we focused on the effect of intestinal abnormalities on the E/I imbalance. Notably, we found that *B. uniformis* partially restored the E/I imbalance at the neural circuit level, suggesting that abnormal synaptic transmission triggered by host defects in the brain remains an important contributor to the E/I imbalance.

We observed that several, but not all, neutral amino acids changed their serum levels in $Chd8^{+/-}$ mice. One possible reason may be due to the fact that not all the neutral amino acid transporters were increased in the intestine. Among the nine neutral amino acid transporters, six of them maintained unchanged expression, while three (*Slc6a19*, *Slc7a8*, and *Slc7a15*) showed increased expression levels. Another possible explanation may be related to the homeostasis of serum amino acid levels.

Besides intestinal amino acid transporters, some other pathways, including intra-organ exchange of amino acids[48] and transport of amino acids from the kidney, are also involved in maintaining serum amino acid levels[49]. Moreover, the transporter preference for some specific amino acids[50] and the concentration of substrates in intestinal contents, may also contribute to the uptake of amino acids in intestines. SLC6A19 is one of the major neutral amino acid transporters in the intestine and absorbs most of the diet amino acids[51]. Our study revealed that the inhibitor of SLC6A19 (S-benzyl-L-cysteine) could reduce serum amino acid levels, which led to decreased amino acid levels in the brain and restored the ASD phenotypes in $Chd8^{+/-}$ mice. However, there is still a possibility that the absorbed S-benzyl-L-cysteine might affect the brain amino acid levels by directly inhibiting the amino acid transporters in the brain. In addition, alternative mechanisms may also exist, such as via vagus nerve and/or neural active microbial metabolites. Future studies are still needed to elucidate the interplay between intestinal amino acid transport and the CNS. Our study suggests that intestinal interventions (e.g., probiotics, inhibitors of intestinal amino acid transporters) may serve as a potential target for development of therapy for ASD.

## Methods

**Mouse maintenance and breeding.** The mice were maintained in the animal facility at the Institute of Microbiology, Chinese Academy of Sciences, under specific pathogen-free conditions. The temperature was kept at 22 ± 1 °C with a relative humidity of 30–70%, and the dark/light cycle was 12 h:12 h (lights on at 7:00 a.m.). The mice were fed with chow diet (SPF-F01-001, SPF Biotechnology Co., Ltd (Beijing, China)). The mice were bred by crossing male heterozygous $Chd8^{+/-}$ mice with female wild-type mice (C57BL/6J). Mice that were crossed for more than three generations were used for experiments. Unless stated

otherwise, male mice or samples from male mice were used in the experiments. After weaning, $Chd8^{+/+}$ and $Chd8^{+/-}$ mice were housed separately by genotype unless stated otherwise.

**Animal ethics statements**. All animal experiments were performed in accordance with the National Institute of Health Guide for the Care and Use of Laboratory Animals. All procedures were approved by the Animal Ethics Committee at the Institute of Zoology, Chinese Academy of Sciences.

**Generation of *Chd8* mutant mice**. The *Chd8* mutant mouse model was established by the Nanjing Biomedical Research Institute of Nanjing University (NBRI, China). Briefly, gRNA was designed to target exon 1 in the mouse *Chd8* gene. Cas9 mRNA and gRNA were injected into mouse zygotes (C57BL/6J). The mice of generation F0 were genotyped by sequencing with a primer specific for exon 1 (*Chd8*-exon 1-R1). The mouse carrying a 13 bp deletion in exon 1 was selected as the founder and was imported to the animal facility at the Institute of Microbiology, Chinese Academy of Sciences. The founder was crossed with wild-type female mice to generate F1 mice. All F1 mice were genotyped by sequencing (*Chd8*-exon1-R1), and male mice carrying the desired mutations were kept and crossed with wild-type female mice to generate F2 mice. F2 and other generations of mice were genotyped with PCR primers specific for the 13 bp deletion allele (*Chd8*-mut-F1/*Chd8*-mut-R1). The sequences of primers were provided in Supplementary Table.

**Behavioral tests**. The test mice were handled for 3 consecutive days before the experiments. All behavioral tests were performed between 8:00 a.m. and 19:00 p.m. in the behavioral test room, where the mice were transferred at least 1 h in advance for acclimation. Male mice aged 8–14 weeks were selected and divided into three cohorts for behavioral tests. Each cohort was used to perform 3–4 behavioral tests. Each behavioral test had results from at least two cohorts.

*Open-field test*. The open-field test was performed in an automated AniLab Eight-Box Locomotor Test Station (AniLab Scientific Instruments Co., Ltd., Ningbo, China). The mouse was introduced to the center of the arena (40 cm × 40 cm) and allowed to explore for 30 min. Total distance traveled, time in the center (24 cm × 24 cm), and movement velocity were recorded using LabState software (AniLab Scientific Instruments Co., Ltd.)[52]. These data were calculated across a total of 30 min and every 10 min (the first 10 min (1st), the second 10 min (2nd), and the last 10 min (3rd)).

*Light/dark box test*. A test mouse was placed in the center of the dark box and allowed to freely explore both chambers for 5 min. The percentage of time spent in the light box was calculated as the index of anxiety-related behavior[52].

*Three-chamber social interaction test*. The three-chamber social interaction test was performed as previously described with minor modifications[53,54]. Briefly, the arena [60 cm (L) × 40 cm (W) × 30 cm (H)] was horizontally separated into three equal-sized chambers by transparent dividers with a square opening in each divider to let the mouse move freely in the three chambers. A stainless cylinder with steel bars was placed in the middle of each left and right chamber. In the first session, the test mouse was introduced to the middle chamber center and moved freely for 10 min. In the second session (social ability session), an age- and sex-matched stranger C57BL/6J mouse (M1) previously habituated to the cylinder was placed in the center of the right chamber, while an object in another cylinder (object) was placed in the center of the left chamber. The test mouse was then allowed to explore freely for 10 min. The movement of the mouse was recorded by a camera over the arena. The interaction time was defined as the time spent sniffing the cylinder when the nose was directly oriented towards the cylinder. The percentages of interaction time in the social ability session were calculated as M1/(M1 + Object) and Object/(M1 + Object), and the preference index was (M1-Object)/(M1 + Object). For the third session (social novelty preference session), another age- and sex-matched stranger C57BL/6J mouse (M2) previously habituated to the cylinder was placed in the center of the right chamber, while M1 was placed in the center of the left chamber. The test mouse was then allowed to explore freely for 10 min. The percentages of interaction time in the social novelty preference session were calculated as M1/(M1 + M2) and M2/(M1 + M2), and the preference index was calculated as (M2 − M1)/(M2 + M1).

*Novel object recognition test*. The novel object recognition test was performed as previously described with slight modifications[55]. Briefly, the mouse was placed in the arena (30 × 40 × 30 cm) to explore two identical objects (circular cone or rectangle pyramid) that were 5 cm away from the sidewall and 25 cm away from each other. Ten min later, the mouse was returned to the home cage. One hour later, one familiar object (Old Object) and one novel object (Novel Object) were placed in the arena where the two identical objects were located previously. Then, the mouse was introduced to the arena to explore for five min. The movement of the mouse was recorded by a camera over the arena. The contact time was defined as the time spent sniffing the object when the nose was directly oriented toward the object. The percentages of interaction time were calculated as Old Object/(Novel Object + Old

Object) and Novel Object/(Novel Object + Old Object), and the preference index was calculated as (Novel Object − Old Object)/(Novel Object + Old Object).

*Reciprocal social interaction test*. The reciprocal social interaction test was performed as previously described with minor alterations[7]. Briefly, a pair of age-, sex-, and body weight-matched unfamiliar mice with the same genotype was placed in the arena (30 × 40 × 25 cm) for 5 min. The movement of the mouse was recorded by a camera over the arena. The time spent in close social interaction (nose-to-nose sniffing, nose-to-anus sniffing, touching, close following, and/or crawling over/under each other) was calculated by a human observer blind to the experimental design.

*Marble-burying test*. The marble-burying test was performed as previously described[18]. The test mouse was placed in an arena containing 20 black marbles sitting on 5 cm of fresh bedding. After 20 min, the mouse was removed, and marbles with at least half of their depth covered were considered buried.

*Self-grooming test*. The self-grooming test was performed as previously described[18]. The test mouse was placed in a clean arena to explore freely for 20 min. Self-grooming behavior was recorded in the second 10 min, as the first 10 min was considered the habituation phase.

*Forced swimming*. The test mouse was placed in a plastic cylinder (height of 30 cm and diameter of 20 cm) filled with water (25 ± 1 °C, 15 cm in depth) and allowed to swim freely for five min. The movement of the mouse was recorded by a camera directly above the cylinder, and the immobility time during the 5 min was calculated by an observer blind to the genotypes. Immobile behavior was defined as motionless floating in the water with little movement[56].

**Metabolomics assay**

*Sample preparation*. Fecal samples were collected, immediately frozen in liquid nitrogen and stored at −80 °C until use. Frozen feces were thawed on ice and resuspended in Milli-Q water (3 μl/mg) on ice. Water extraction of feces was performed by centrifugation (4 °C, 15871 rcf, 20 min), and the sample was stored at −80 °C. Fifty microliters of fecal water was removed, and 150 μl of methanol was added. The mixture was vortexed and stored at 4 °C for 1 h before centrifugation (4 °C, 15871 rcf, 20 min). The fecal samples for the metabolomics assay were obtained and stored at −80 °C until use.

The brain samples were harvested, immediately frozen in liquid nitrogen and stored at −80 °C until use. The brain tissues were homogenized in methanol (10 μl/mg). After centrifugation (4 °C, 15871 rcf, 20 min), the supernatant was stored as the brain sample for the metabolomics assay at −80 °C until use[57].

Blood was collected from the mouse eyes and was stored undisturbed at room temperature for 20 min before centrifugation (2000 rcf, 10 min). Fifty microliters of supernatant was removed, and 150 μl of methanol was added. The mixture was vortexed and stored at 4 °C for 1 h before centrifugation (4 °C, 15871 rcf, 20 min). The serum sample for metabolomics was obtained and stored at −80 °C until use.

*Targeted metabolomics assay for amino acids and SCFAs*. Metabolomics assays were performed on an ABSciex QTRAP 6500 mass spectrometer interfaced with a Waters ACQUITY UPLC equipped with a BEH C18 column (2.1 × 100 mm, 1.7 μm; Waters). Amino acids and SCFAs were monitored in multiple reaction monitoring modes using characteristic parent-daughter ion transitions at $m/z$ ratios for each amino acid and SCFA. Several concentrations of amino acids and SCFA standards were used for quantification.

**mEPSC and mIPSC recording**. mEPSC and mIPSC recordings were performed as previously described with minor modifications[18,58]. Briefly, adult mice (8–9 weeks) were anesthetized by chloral hydrate and transcardially perfused with aCSF containing: 185 mM Sucrose, 2.5 mM KCl, 1.2 mM $NaH_2PO_4$, 25 mM $NaHCO_3$, 25 mM D-Glucose, 0.5 mM $CaCl_2$, and 10 mM $MgSO_4$. After decapitation, the brain was quickly removed, and coronal slices (300 μm) of the medial prefrontal cortex were prepared. The slices were incubated in aCSF for 1 h before mEPSC and mIPSC recording. All solutions were bubbled with 95% $O_2$ and 5% $CO_2$.

Pyramidal cells of medial prefrontal cortex layer V were visualized under an IR-DIC microscope. Whole-cell recordings were obtained using a patch electrode filled with intracellular solution (for mEPSCs, 140 mM K-gluconate, 2 mM $MgCl_2$, 10 mM HEPES, 8 mM KCl, 2 mM $Na_2$-ATP, 0.2 mM NaGTP, pH 7.3; for mIPSCs, 140 mM CsCl, 2 mM $MgCl_2$, 10 mM HEPES, 1.1 mM EGTA, 2 mM $Na_2$-ATP, 0.1 mM CaCl, pH 7.3). Perfusion solution containing 1 μM TTX, 10 μM NBQX, and 50 μM DAP5 was used to block excitatory synaptic transmission in mIPSC recordings. In the mEPSC recordings, 1 μM TTX, 10 μM bicuculline, and 50 μM DAP5 were added in aCSF. The membrane potential of pyramidal neurons was held at −70 mV.

***B. uniformis* administration**. *B. uniformis* D2-69 was grown anaerobically on Gifu Anaerobic Broth (GAM Broth) in anaerobic tubes at 37 °C. For gavage, the bacteria were centrifuged in an anaerobic tube and suspended in anaerobic PBS to a

concentration of $5 \times 10^8$ colony forming units (CFU)/ml[59]. The *B. uniformis* suspension was removed from the anaerobic tube and immediately administered to the mice orally by gavage (0.1 ml for mice aged 4–5 weeks; 0.2 ml for mice older than 5 weeks).

**GM6001 administration**. GM6001 (Abcam, Cat # ab120845) was dissolved in DMSO (10 mg/ml) as the stock solution and stored at −80 °C until use. Before administrating to mice, the stock or DMSO was further diluted with four times 10% Tween 20 (v/v, diluted in normal saline) to make a final DMSO concentration of 20%. Adult mice (8–9 weeks) were intraperitoneally injected with diluted GM6001 (5 ml/kg BW) or diluted DMSO (5 ml/kg BW) every day for 5 consecutive days[60,61]. After the last injection, feces were collected before the mice were sacrificed.

**Drug administration**. For glutamine administration, glutamine (13 mg/ml or 25 mg/ml) was supplemented in the drinking water of the mice[62]. For S-benzyl-L-cysteine administration, it was dissolved in water at a concentration of 20 mmol/l. NaOH was added to help dissolve the chemical, and the pH was adjusted to 8.5[63]. For nimesulide administration, suspensions of nimesulide were prepared in 0.05% carboxy methyl cellulose at a concentration of 0.65 mg/ml[64]. The drug solutions were sterilized with a 0.22 μm membrane before being given to the mice, and the solutions were changed twice a week.

**Fecal microbiota transplantation**. Feces from $Chd8^{+/+}$ and $Chd8^{+/−}$ mice have been collected immediately into 3 ml cryoprotectant buffer (normal saline containing 30% glycerin and 0.1% L-cysteine) to the final volume of 4 ml. The feces were dispersed in the cryoprotectant buffer anaerobically and then passed through a 70 μm cell strainer anaerobically. The filtrate was stored at −80 °C until use.

Germ-free mice were separated into two groups according to their body weight (Day 0) and gavaged with fecal filtrate from $Chd8^{+/+}$ and $Chd8^{+/−}$ mice separately three times (Day 0, Day 1, Day 3). Two weeks after the first gavage (Day 14), the mice were sacrificed, and their tissues were obtained.

**Enzyme-linked immunosorbent assay (ELISA)**. The small intestinal contents, cecal contents, and colonic contents were harvested and suspended in Milli-Q water (3 μl/mg) on ice before centrifugation (4 °C, 15871 rcf, 20 min). The supernatant was used for defensin α1 and defensin α2 detection using ELISA kits (JL48732 and JL48735, Jianglaibio, Shanghai, China) according to the manufacturer's instructions.

**Immunofluorescence**. For preparation of intestinal cross sections, mouse ileal segments were harvested, and the debris in the lumen was flushed with ice-cold EBSS (SL6160, Coolaber, China). Tissue sections were fixed in 4% paraformaldehyde (PFA) for 1 h before dehydration for 2 h each in 20% and 30% sucrose. Dehydrated tissues were frozen in OCT and cut 10 μm thick using a cryostat (Leica CM1950, Germany).

For preparation of coronal sections of the brain, a mouse was transcardially perfused with normal saline followed by 4% PFA immediately. The brain was removed and fixed with 4% PFA overnight before dehydration for 24 h in 20% and 30% sucrose. Dehydrated tissues were frozen in OCT and cut 30 μm thick using a cryostat (Leica CM1950, Germany).

For immunofluorescence staining, brain or ileal sections were permeabilized with 0.3% Triton in 5% BSA (V900933, Sigma, USA) diluted in PBS for 1 h at room temperature for staining. The permeabilized sections were incubated with primary antibodies: anti-SLC6A19 (1:1000, ab180516, Abcam), anti-SLC7A8 (1:1000, ab75610, Abcam), or anti-oxytocin (1:2000, 20068, ImmunoStar) before washing and incubation with secondary antibodies (1:500, A-11012, Thermo Fisher Scientific) and DAPI (C1002, Beyotime, Haimen, China). The sections were washed again and mounted with Fluoromount-G (0100-01, Southern Biotech, USA) on slides with coverslips. The slides were stored in the dark at 4 °C until they were imaged using a spinning disk confocal super-resolution microscope (SpinSR10, Olympus, Tokyo, Japan). ImageJ software (NIH) was used for image processing and quantification.

**Western blot**. Tissues were homogenized in lysis buffer (P0013B, Beyotime, Haimen, China) containing protease inhibitor cocktail (P0013B, Beyotime, Haimen, China). The homogenates were centrifuged (4 °C, 15871 rcf, 20 min), and the supernatant was collected. Approximately 20 μg of proteins was loaded on a Tris-glycine gel. Proteins were transferred onto PVDF membranes, blocked, and stained overnight using the following primary antibodies: anti-CHD8 (1:1000, ab114126, Abcam), HRP-conjugated anti-GAPDH (1:5000, BE0034, Easybio, Beijing, China), anti-SLC6A19 (1:5000, ab180516, Abcam), and anti-SLC7A8 (1:5000, ab75610, Abcam). Membranes were washed and stained with HRP-conjugated secondary antibodies (1:5000, BE0101 and BE0102, Easybio, Beijing, China) before washing and development with enhanced chemiluminescence (ECL) substrate for Western blotting (BE6706, Easybio, Beijing, China).

**Bulk RNA-seq**. Total RNA was extracted from the small intestines of 12-week-old mice using TRIzol (Invitrogen) according to the manufacturer's instructions. There were three, three and five mice in the KN, KB and WN groups, respectively (WN: $Chd8^{+/+}$ mice gavaged with PBS; KN: $Chd8^{+/−}$ mice gavaged with PBS; KB: $Chd8^{+/−}$ mice gavaged with *B. uniformis*). mRNA sequencing libraries were prepared and sequenced on the NovaSeq6000 platform (Illumina), generating an average of 20 million paired-end reads per sample.

**scRNA-seq**. Adult mice (12 weeks) were anesthetized and decapitated, and the brain and small intestine were removed immediately and submerged in ice-cold aCSF and D-HANKS, respectively. Single cells from the brain were harvested using an Adult Brain Dissociation Kit (130–107-677, Miltenyi Biotec, Germany) according to the manufacturer's instructions. Single cells from the small intestine were obtained by digestion with papain (0.8 U/ml, Sigma, P3125), DNase I (50 U/ml, Coolaber, CD4871), FBS (20 μl/ml, Gibco), collagenase I (31.25 U/ml, Sigma, C0130), and collagenase IV (31.25 U/ml, Sigma, C5138). Single-cell suspensions of freshly isolated cells were then resuspended in PBS containing 0.04% ultrapure BSA. scRNA-seq libraries were prepared using Chromium Single-Cell 3' Reagent Kits v2 (10× Genomics) according to the manufacturer's instructions. The target cell recovery for each library was 7000. Generated libraries were sequenced on an Illumina HiSeq2500 system (Beijing Institutes of Life Science, Chinese Academy of Science).

**Whole-genome shotgun metagenomic sequencing and 16S rRNA sequencing**. Microbial DNA was extracted from the frozen fecal samples using a TIANamp Stool DNA kit (DP328, Tiangen Biotech, Beijing, China) according to the manufacturer's instructions. The DNA quality and concentration were measured by agarose gel electrophoresis and a Qubit fluorometer (Life Technologies, Waltham, MA) before downstream high-throughput sequencing.

To explore the gut microbiota of the $Chd8^{+/+}$ and $Chd8^{+/−}$ mice, we performed metagenomic sequencing on frozen fecal samples. There were 11 $Chd8^{+/−}$ mice and 12 $Chd8^{+/+}$ mice in each group. Extracted DNA was fragmented and used to construct a genomic library with an average insert size of ~280 bp and then sequenced by NovaSeq6000 (Illumina).

To study the effect of the increased defensin α1 expression on the gut microbiota, we performed 16S rRNA sequencing on frozen fecal samples from the $Chd8^{+/+}$ mice gavaged with defensin α1 or water. There were ten mice in each group. For 16S rRNA sequencing, variable regions 3 and 4 (V3–V4) of the 16S rRNA gene were amplified using the 241F and 805R primers. The purified V3–V4 amplicons were sequenced on a HiSeq2500 platform (Illumina, San Diego, CA) with paired-end 250 bp sequencing.

**Transcriptome and metagenome analyses**

*scRNA-seq data analysis*. Data produced by scRNA-seq were processed with Cell Ranger (version 3.1.0) software[65]. In detail, this program was used to demultiplex the FASTQ reads and align them to the mouse reference genome (mm10) with default parameters. The result contains a digital gene expression (DEG) matrix for each sample, with the number of unique molecular identifiers (UMIs) collapsing for each gene. Next, the DEG matrix was subjected to Scanpy (version 1.6.0)[66]. Quality control was performed, and cells with more than 6000 or less than 200 detected genes and cells with a mitochondrial transcription ratio >5% were discarded. In addition, cell doublets were discarded after identification using the scrublet tool. After $\log(x + 1)$-transformation, size factors were estimated using Scran v1.16.0, and normalization was performed[67]. Data were then removed based on batch effects and biological covariates with Harmony (version 0.0.5)[68]. Next, cells were divided into unsupervised clusters using graph-based clustering of the PCA reduced data with the Louvain algorithm after computing the nearest neighbor graph to partition the cells. The results were visualized by uniform approximation and projection (UMAP, version 0.4.6). Finally, the resulting coordinates of UMAP and cluster tags for each cell were assigned to the expression matrix after the size factor correction step for downstream analysis. DE genes between groups in each cluster were detected using the Wilcoxon rank-sum test in diffxpy.api in Scanpy. Adjusted $P$ values were separately evaluated for each cell subset comparison using the Benjamini–Hochberg correction. Finally, cells were partitioned into epithelial, stromal, neuronal, endothelial and immune compartments based on the expression of canonical cell type-specific markers.

*Bulk RNA-seq data analysis*. For bulk RNA-seq data, transcripts were mapped and quantified using the HISAT2 (version 2.0.5)-StringTie (version 1.3.4) pipeline[69]. A total of 20,413 genes supported by more than five reads in 20% of the samples were selected for downstream analysis. Differential expression analysis was performed by DESeq2 (version 1.24.0)[70]. There were 560 and 475 DE genes in KB_WN and KN_WN, respectively, under a strict threshold of adjusted $P < 0.01$ and $|\log 2(cf)| > 0.585$, and then, these DE genes were used in Gene Ontology term enrichment analysis (GOEA). A total of 13,611 highly expressed genes with an average FPKM greater than 1 were used in gene set enrichment analysis (GSEA). GSEA and GOEA were performed with clusterProfiler (version 3.12.0)[71]. The results of GOEA were visualized in GOplot (version 1.0.2). Euclidean distance was calculated based on the FPKM of DE genes in the three groups to compare the

function dist (method = "euclidean") in R. The Mann–Whitney test was performed in R to compare the difference in Euclidean distance between different groups.

*Taxonomic analysis.* For metagenomic sequencing data, cutadapt (version 1.14) was used for adaptor and quality trimming. Mouse DNA was removed using BowTie2 (version 2.3.2) to map the reads against the mouse reference genome. Ten million reads were randomly selected from each sample to ensure the consistency of data size in different samples. Quantitative taxonomic profiling was carried out using DIAMOND (version 0.9.24) by aligning reads to the NR database with the −f 102 parameter[72,73].

For 16S rRNA-seq data, raw sequencing data were analyzed using QIIME2 (version: 2020.2.0)[74]. First, raw reads were denoised, dereplicated, merged, and finally processed by DADA2 in QIIME2[75]. For taxonomy annotation, following the recommendations of the QIIME2 (https://docs.qiime2.org/2021.8/tutorials/overview/), we used the blast-based classification method (classify-consensus-blast) to compare the representative sequence to a reference database of sequences with known taxonomic composition (Greengenes 99% OTUs database (version:13_8). The microbial profile table was then exported as a text file for subsequent analyses.

For diversity analysis, first the microbial table was normalized by total sum reads with the function of *transform(transform = "compositional")* in the R package microbiome (version 1.6.0). Alpha diversity was analyzed by function *diversity* in the R package vegan (version 2.5.6), and statistical significance was calculated by the Mann–Whitney test in R. For beta analysis, principal component analysis (PCA) was performed by the *prcomp* function in R and was visualized with the R package ggbioplot (version 0.55). Multivariate analysis of variance using distance matrices (PERMANOVA) was performed using the *adonis(permutations = 9999,method = "bray")* function in the vegan package. Analysis of similarities test (ANOSIM) was also used to detect the degree of separation between groups by the *anosim(permutations = 9999,distance = "bray")* function in the vegan package. The BC distances of the two groups of samples were also compared by Mann–Whitney test. And we also reanalyzed the beta diversity and distance based on the centered log-ratio transformed Euclidean distances by using *transform(transform = "clr")*. We obtained results with altered significance but consistent trends.

For differential analysis of metagenomic data, the highly reliable microbiota supported by more than 10 reads in 10% of the samples was selected by the Mann–Whitney test in R. All *P* values were adjusted for multiple testing by the Benjamini–Hochberg approach by the function *p.adjust(method = "BH")* in R. DESeq2 was used in the differential analysis of the 16S rRNA data. Discriminant microbial features between the study groups were determined using linear discriminant analysis (LDA) effect size (LEfSe) on the Galaxy platform (https://galaxy.medunigraz.at, *P* < 0.05, LDA score >3.0).

*Functional analysis of metagenomic data.* Functional profiling was performed with HUMAnN2 (version 0.11.1)[76], which aligned reads to selected databases (KEGG, GO and COG) using BLAST or other alignment tools to characterize functional profiles. In total, we identified 5220 KOs, and to facilitate comparisons between groups, we normalized the abundance of KOs to copies per million (CPM). Differential abundance analysis was carried out on the 2781 KOs with a mean CPM > 1. Then, 301 KOs (Mann–Whitney test, *P* < 0.05) were used for functional enrichment analysis in the R package clusterProfiler. The relationship between KEGG pathways and KOs was downloaded from the KEGG database. The result was visualized in the R package GOplot (version 1.0.2).

*qPCR analysis.* Total RNA isolated from mouse intestines was used for cDNA preparation using a Hifair® II 1st Strand cDNA Synthesis Kit (11121ES60, Yeasen, Shanghai, China). Real-time PCR analysis was performed using Hieff® qPCR SYBR Green Master Mix (11203ES03, Yeasen, Shanghai, China) in a StepOne PLus Real-Time PCR system instrument (Applied Biosystems) to determine the expression levels of defensin genes (*Defa1, Defa2, Defa4, Reg3a, Reg3b, Reg3g, Lyz1, Defb3*), amino acid transporter genes (*Slc6a19, Slc7a8, Slc7a15*), intestinal tight junction genes (*Ocln, Cldn1, Tjp1*), and the *Gapdh* gene. The sequences of primers were provided in Supplementary Table 1.

*Intestinal permeability assay.* The assay was based on intestinal permeability towards 4 kDa fluorescein isothiocyanate-conjugated dextran (FITC-dextran) as previously described[77]. Briefly, the mice were fasted for 4 h and injected with FITC-dextran (600 mg/kg body weight, 80 mg/ml in PBS) by gavage. The mice were sacrificed 4 h later, and FITC-dextran concentration in the serum was determined with a fluorescence spectrophotometer (excitation at 485 nm and emission at 535 nm; Infinite 200 PRO, Tecan) according to the standard curve.

*Blood–brain barrier (BBB) integrity detection.* Adult mice (8–9 weeks) were administrated Evans blue dye (2% w/v; intraperitoneal injection; 10 ml/kg BW) in normal saline. Two hours later, mice were transcardially perfused with ice-cold normal saline immediately and brain obtained. Cortex was removed and placed in 50% (w/v, in distilled water) ice-cold trichloroacetic acid (1 ml/0.2 g tissue) before homogenization. The homogenates were incubated at 4°C for 30 min, followed by centrifugation (4 °C, 15871 rcf, 20 min). Evans blue concentration in the supernatant was determined

with a fluorescence spectrophotometer (excitation at 540 nm and emission at 680 nm; Infinite 200 PRO, Tecan) according to the standard curve.

*Assessment of feeding behavior.* Adult mice (11 weeks) were housed in groups of two to four animals per cage. Body weight and food consumption were monitored once a week over 6 weeks. Weekly body weight was shown as the average value during the week. The food intake of mice was calculated as the weight of food pellets consumed per kg body weight for each cage[78].

*Gastrointestinal transit test.* Adult mice were injected with 200 μl of charcoal marker (10% charcoal and 5% gum arabic dissolved in water) by gavage after food deprivation for 12 h with free access to water. The mice were sacrificed by cervical dislocation 30 min later, and the intestine from the pyloric junction to the ileocecal junction was removed for measurement of the total length of the intestine and the distance traveled by the charcoal marker. Gastrointestinal transit is expressed as the distance traveled by the charcoal marker and the percentage of distance traveled by the charcoal marker relative to the total length of the small intestine[15,79].

*Multiplexed determination of cytokines.* Colon samples were harvested and frozen in liquid nitrogen immediately and stored at −80 °C until use. Colon samples were homogenized in lysis buffer (P0013B, Beyotime, Haimen, China) containing protease inhibitor cocktail (P0013B, Beyotime, Haimen, China) on ice for protein extraction. The homogenates were centrifuged (4 °C, 15871 rcf, 20 min), and the supernatant was collected as colonic protein samples. The serum samples were obtained and stored as described above. The levels of eight cytokines of interest (IL-1β, IL-4, IL-6, IL-22, IL-10, IL-12p70, IL-17A, and IL-33) in the serum and colon were measured using a personalized Milliplex Map kit (Millipore) following the manufacturer's instructions. The concentrations of cytokines in the colon were normalized to the protein concentrations of the samples.

**Reporting summary.** Further information on research design is available in the Nature Research Reporting Summary linked to this article.

## Data availability
The raw sequence data reported in this study have been deposited in the Genome Sequence Archive (GSA) in National Genomics Data Center, Beijing Institute of Genomics (China National Center for Bioinformation), Chinese Academy of Sciences under the accession code PRJCA007907. In detail, the 16S rRNA and metagenomic sequencing data have been deposited under the accession code CRA005819. The bulk RNA-seq data have been deposited under the accession code CRA005826. The scRNA-sequencing data have been deposited under the accession code CRA005899. The remaining data generated in this study are provided in the Supplementary Information. Source data are provided with this paper.

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

## Acknowledgements

This work was supported by grants from National Natural Science Foundation of China (32025009, 32001082), the National Key R&D Program (2021YFA1301000, 2021YFC2300017), and the Strategic Priority Research Program of Chinese Academy of Sciences (XDB38020300).

## Author contributions

F.Z. conceived the project. Y.Y., Z.Z., Y.H., C.L., and S.L. performed the experiments and generated the sequencing data. Y.Y., B.Z., and P.J. performed the data analysis. Y.Y. and F.Z. wrote the paper with the contribution of all authors.

## Competing interests

The authors declare no competing interests.
