## [Peer Review File · Nature Communications]

Reviewers' Comments:

Reviewer #1:

Remarks to the Author:

Yu and colleagues examined the role of the gut-microbiota-brain axis in a *Chd8*-deficient mouse model for Autism Spectrum Disorder (ASD). The authors found that in addition to anxiety-like behavior and deficits in social behavior and memory, *Chd8*^{+/-} mice also show changes in their gut microbiota composition, including reduced levels of *B. uniformis*. Moreover, the mutant mice also exhibit changes in intestinal amino acid transporter levels and in the circulating levels of amino acids such as glutamine. Interestingly, treatment with *B. uniformis* rescues selective behavioral deficits, notably the preference for social novelty and anxiety-related behavior but not the memory deficits. *B. uniformis* also rescues changes in the levels of amino acid transporters in the intestines and in the E/I balance in the brain. Moreover, pharmacological inhibition of amino acid transporters rescued the social deficits and anxiety-related behaviors.

The authors provide compelling evidence that the gut-microbiome, specifically *B. uniformis*, can improve some, but not all, behavioral symptoms associated with the partial loss of *Cdh8*. However, there are a few areas of the manuscript which are necessary to address in order to strengthen the work as a whole. Please see below for specific comments:

Major comments/concerns:

1. The statistics for many of the behavioral and molecular experiments are not correct. Mann-Whitney U tests should only be used when comparing two groups or variables. However, the authors use it for comparisons of more than two.
 - a. For example, in Figure 1b, f, I, a two-way ANOVA with appropriate post-hoc correction (e.g., Bonferroni) should be used given that there are two variables per group.
 - b. Additionally, for example, in Figure 5a, c, e, and f (but in many other figures as well), a one-way ANOVA with appropriate post-hoc correction (e.g., Tukey or Bonferroni), or a comparable non-parametric analysis such as Kruskal-Wallis test, should be used given that there are 3 groups.
2. For the microbiome-related research, it is important to provide more details about the housing conditions. In particular, the authors should state whether the *Chd8*^{+/+} and *Chd8*^{+/-} mice were housed together or separated by genotype after weaning given recent reports that such conditions may impact both the microbiome and behavior of animals. If the *Chd8*^{+/-} and *Chd8*^{+/+} mice are housed separately, the authors should co-house these mice to determine whether the behavioral deficits are purely microbial driven.
3. It is not clear whether the increased levels of the amino acid transporters are due to the genetic mutation or the changes in the microbiome. To answer this question, the authors could transfer the microbiome from the WT and mutant mice into germ-free (GF) mice. Alternatively, the authors could make the *Chd8*^{+/-} mice GF and measure levels of amino acid transporters in the GF-*Chd8*^{+/-} mice.
4. Are there any changes in amino acid transporters in the brain or blood brain barrier permeability in *Chd8*^{+/-} mice? Moreover, does *B. uniformis* treatment affect brain amino acid transporter levels? Previously, changes in the levels of amino acid transporters in the blood brain barrier have been shown to lead behavioral symptoms of ASD (Tărlungeanu et al. Cell 2016). Moreover, the authors do not show whether the inhibitory effect of S-Benzyl-L-cysteine could be working in the brain.
5. Is *Chd8* expressed in immune cells? Does loss of *Chd8* in immune cells drive inflammation in this model?
6. The authors' claim that increased levels of alpha defensins contribute to gut microbiota dysbiosis in *Chd8*^{+/-} is not supported by the evidence shown. To support this, the authors need to test whether inhibition of alpha defensins in *Chd8*^{+/-} mice normalizes the microbiome.
7. Did the authors look at behavior of the defensin alpha 1 treated mice? Is increased defensin alpha 1 sufficient to cause behavioral deficits?

8. Does treatment with *B. uniformis* rescue the changes in the gut morphology, gut microbiota, and inflammatory markers of the *Chd8*^{+/-} mice which the authors describe in figures 1 and 3?

9. The authors claim that the increase in serum L-glutamine drives the social deficits and anxiety-related behaviors in the *Chd8*^{+/-}. However, the data supporting this are correlational. Does treatment with L-glutamine cause social deficits and anxiety-like behavior in WT mice?

10. For the microbiome data, the authors should use PhILR-transformed or CLR-transformed Euclidean distances in addition bray-curtis distances when measuring beta-diversity and performing ordination. Additionally, the authors should consider using ASV outputs instead of the 99% OTUs for looking at taxonomy from the 16S data in order to match the standards methods of the field.

Minor comments:

- On page 2 paragraph 2, the authors state "A recent study found that the gut microbiota from ASD patients is sufficient to trigger the disease in healthy individuals" citing the work from Sharon et al., Cell 2019. The statement is a mischaracterization of the findings and the overall results/interpretations of the FMT experiments from this paper are highly controversial.

- For the open field test, "time in center" is a preferred measurement of anxiety-like behavior compared to "distance travelled in center." The authors should consider showing data for "time in center" as well.

- On page 4 paragraph 3, the authors say that they examine "region-specific" changes in amino acids by comparing levels in the cerebrum and brain stem. However, this is not what is typically considered region-specific. Instead, examining changes in areas within the cerebrum such as cortex, hippocampus, hypothalamus, etc. is what is considered region specific. Given that the mEPSCs and mIPSCs were recorded in cortical neurons, it would make sense to look for region-specific changes in amino acids in the cortex as well.

- On page 2, paragraph 3, the authors claim that their findings represent "a new mode of communication of the gut-brain axis is ASD" distinct from "neuroactive compounds." However, changes in amino acids in the brain due to the microbiome-mediated changes in circulating amino acids have already been shown to underly the anti-epileptic effects of the ketogenic diet/probiotic species (Olson et al., Cell 2018) and changes in the levels of other metabolites such as 5-aminovaleic acid has also been shown to improve some ASD-related behaviors (Sharon et al., Cell 2019). Moreover, changes in brain amino acids due to changes in amino acid transporters has been shown in ASD (Tărlungeanu et al. Cell 2016). While the data the authors show are novel, they do not necessarily constitute an entirely new gut-microbiome-brain pathway as they claim.

- On page 3, paragraph 3: the phrase "non-significantly different social abilities" is not very clear/fluent. The authors should consider describing the result as "normal sociability" or "normal preference for social over non-social interaction."

- On page 3, paragraph 2: While there is a clear trend in intestinal transit time data (Ext Data Fig 1e), the difference between groups is not statistically significant. Therefore, the authors should consider rephrasing "slightly decreased intestinal motility" as "a trend towards decreased intestinal motility" in order to more accurately convey the results.

Reviewer #2:

Remarks to the Author:

Yu et al present a very interesting manuscript entitled *Bacteroides uniformis* restores ASD-like phenotypes by reducing intestinal amino acid transport. In this study, the authors show evidence that *CHD8* het mutants display abnormal behaviors, show changes in the gut and in gut flora, and that these microbiota changes in the gut might mechanistically result in the changes in behavioral phenotypes. The study utilizes an array of approaches and reveals several novel findings further

supporting a critical role for the gut-brain axis in the regulation of neuro-psychiatric relevant behaviors. Enthusiasm is somewhat lessened by a number of concerns detailed below regarding the proposed mechanisms. Specifically, the mechanism proposed is that AA transporters are altered in the gut, resulting in increased serum glutamine, that results in increased brain glutamine and then glutamate and resulting E/I imbalance. However, in Figures 6c-d, serum glutamine does not correlate necessarily with brain glutamine. Also, as brain glutamine is significantly, if not mostly, contributed by astrocytic production of glutamine, it will be important to show greater evidence that the proposed mechanism is the driver of the E/I imbalance observed in these animals. Also, glutamine levels in various experiments are quite variable. Overlapping levels are both associated with normal behaviors and with abnormal behaviors, which further raises questions about the proposed mechanism. Also, statistical measures do not seem ideally utilized for the studies performed. The authors should also be cautioned not to define some of the behaviors examined as being equivalent to anxiety or depression: depression- or anxiety-like perhaps would be more appropriate in this context.

Specific Comments

Figure 1. Please include size markers for westerns. It is also not clear why the authors check hypothalamus in adult when E14.5 is whole brain? Also, What is level of 110kDa isoform at E14.5?

1g: For behavior, the authors plot distance in center. What about time in center (which would probably be more comparable with duration in light box of light dark box). Also, authors should show locomotor activity in OF across time bins (not just total)?

The authors should at least discuss how much of behavior may relate to increased anxiety-like behavior?

The authors show that mutants possess smaller gut (small and large intestine), decreased motility, and decreased weight. How do these factors relate/interact with the subsequent findings re: intestinal flora, absorption, transporters?

Figure 2: In figure 2 g,h: how is quantification of expression performed? From plots, it appears that more slc__ expressing cells are present. It's not clear if the intensity of the color scheme denotes level of expression or not, but it's not clear that level of expression vs. number of cells expressing the transporters has been altered. In immuno's in 2i, the figure suggests a massive increase in transporter levels (almost all or nothing, especially for 6A19). However, in the quantification, it is merely < 1.5fold increase. The figure does not appear to be representative. Re: metabolic studies: to my understanding, glutamine is mostly produced from muscle for use in the body while most of brain glutamine is locally synthesized in astrocytes. It will be important to show that in fact brain levels are directly impacted by these gut transporters (the gavage experiments and the transporter blocker experiments are consistent with the hypothesis but aren't wholly convincing that alternative mechanisms aren't at play. Also, slc6a19 transports a number of AA's including glutamine, leucine, isoleucine, meth. However, only glut is increased. Why do the authors feel that a change in the transporter would result in a specific increase? Also, for Slc7a8, alanine, serine, threonine, cysteine, phenylalanine, tyrosine, leucine, arginine and tryptophan should be transported. Again, of these, the only AA that is increased is tryptophan. This mechanism should either be discussed at minimum or experimentally evaluated.

Fig3. The Chd8^{+/-} mice showed a decrease in the abundance of Bacteroides in the gut microbiota, but this decrease was not found in the defensin α 1-treated mice (Extended Data Fig. 4j). If this is the proposed mechanism by which bacteroides is decreased in mutant mice, shouldn't a decrease be reproduced? Also when you infuse defensin, firmicutes goes up (this is not a species examined/observed in 3b)? In essence, the change in bacterial flora (mutant vs. defensin) have some overlap but are quite distinct.

Fig4. With *B. uniformis* gavage, I may have missed this, but did this procedure actually increase *B. uniformis* abundance, especially relative to other species? Does this manipulation impact the overall gut flora? Also, is the observed effect specific to *B. uniformis*? If you infuse another species, firmicutes for instance which goes up when you put in defensin, do you observe a similar impact ?

4g: Why are ratios of metabolites so different from figure 2 (glu/gaba and levels of glutamine)? Mutant levels are lower here than in controls in fig2. If levels/abundance are the critical measure to generate E/I imbalance, why is this lower level still associated with E/I imbalance and why is it

not imbalanced in controls in 2 when the glutamine levels are much higher?

4f: mIPSC appears unchanged. Do authors propose that increase in mEPSC freq is the driver of the E/I phenotype?

For this figure, it will be important to show locomotor behaviors. The authors state that their findings show "increased anxiety" 4c-d. I think the data might argue the opposite. Authors also use the word restore which may not be the proper term when improving behavior. Authors should also be cautioned about these behaviors equating to anxiety or depression. They are at best a test for anxiety-like behaviors.

Fig5: again, like in figure 4, in mutant, serum levels of glutamine here are comparable to those in controls in Figure 2. Yet, here they are abnormal, but there, normal.

5e: Like with Figure 2, fluorescence intensity here for slc6a19 is quantified as less than a 2 fold increase, yet staining image is from 0 to extremely bright. Either this is not a fair representative increase or the graphs are not representative. In the individual values, the greatest increase is 3 fold. Again, does not match image. Similarly, in f, western for slc6a19 increase is higher in bar graph but appears to be less increased than increased for slc7a8 in blot shown)

Also, as with defensin gavage studies, AA changed with b uniformis gavage are all very different than changed in mutant. Glutamine is changed, although it's not clear that it's significantly changed from control. Also, tryptophan, which was the other AA changed in mutant, is not increased while others appear to be changed that weren't changed in mutant.

This is relevant to the entire study, the authors use a Mann-whitney test here. It is most likely not the most appropriate test to use. This test should be used for comparison of 2 populations. For all comparisons here, there are three populations to be compared. This applies to many of the figures and to all social interaction figures which should not only have comparison between novel animal and familiar animal, but also, how that differs from control.

Figure 6: What is function of s benzyl L cysteine? Authors do not show proof that it inhibits AA transporter, only that it reduced serum glutamine. Also, does it impact anything else or is its actions specific to the transporter? Also, administration does not decrease brain levels despite decrease in serum. So if functions at the intestine only, then why does it impact mutant but not control (i vs. d). and if the proposed mechanism that brain follows serum, it's not clear why this disconnect is present. Also, why do authors evaluate cerebrum in one place and brain in another? Here, glutamine levels in controls are higher than glutamine levels of mutant in fig2 and 4. Again, why do these animals not have E/I imbalance or abnormalities? Also, what is E/I imbalance/electrophysiology in this paradigm? Again, stats need to be performed correctly.

Reviewer #3:

Remarks to the Author:

In this work, Yu et al investigate the effect of the chromatin remodeler Chd8 on interactions between the microbiome and the host, focusing on effects related to ASD and gastrointestinal functions. The authors perform an in-depth study, establishing effects on a wide number of behavioural, neurological and physiological effects, as well as effects on microbiome composition and function.

I am quite excited about this study! The study is both novel and significant for the field, as it establishes mechanistic relationships between host genotype, the microbiome and host phenotypes. This is exactly the kind of detailed work that the microbiome field needs, as there currently is a large amount of associative data, but we are only scraping the surface in terms of cause-effects relations in terms of the effect that the microbiome has on host physiology.

The study is very thorough, and mostly supports its claims through multiple assays and rigorous analysis using appropriate methods. The data analysis is thorough and sound. The manuscript is overall very well-written.

Comments:

1. The main issue I have with the study is the claims regarding defensin-1. The data presented

does not support that changes in microbiome composition require defensin-1 and are not significantly different. It is also unclear why not all conditions are analysed in parallel and shown in the same panel. I suggest the data can be shown but that the claim is removed – overall the study is so informative that the overall conclusions will not be affected.

2. The authors do not seem to have measured if feeding behaviour is affected by in Chd8+/- . A change in feeding could explain many if not all of the phenotypes so needs to be measured.

3. p. 3: 'The Chd8+/- mice exhibited non-significantly different social abilities': I found this sentence confusing – can it be made clearer what the authors mean? Text needs to be clearer to explain that there are behavioural differences, as written now it sounds like they observe differences that are not significant.

4. p. 7 'These results indicated that the increased levels of defensins and inflammatory factors in the intestine may jointly lead to dysbiosis of the gut microbiota in the Chd8+/- mice.': The authors claim changes in cytokines affect microbial composition, when it could be the other way around. Please change the claim.

REVIEWER COMMENTS

Reviewer #1 (Remarks to the Author):

Yu and colleagues examined the role of the gut-microbiota-brain axis in a *Chd8*-deficient mouse model for Autism Spectrum Disorder (ASD). The authors found that in addition to anxiety-like behavior and deficits in social behavior and memory, *Chd8*^{+/-} mice also show changes in their gut microbiota composition, including reduced levels of *B. uniformis*. Moreover, the mutant mice also exhibit changes in intestinal amino acid transporter levels and in the circulating levels of amino acids such as glutamine. Interestingly, treatment with *B. uniformis* rescues selective behavioral deficits, notably the preference for social novelty and anxiety-related behavior but not the memory deficits. *B. uniformis* also rescues changes in the levels of amino acid transporters in the intestines and in the E/I balance in the brain. Moreover, pharmacological inhibition of amino acid transporters rescued the social deficits and anxiety-related behaviors.

The authors provide compelling evidence that the gut-microbiome, specifically *B. uniformis*, can improve some, but not all, behavioral symptoms associated with the partial loss of *Cdh8*. However, there are a few areas of the manuscript which are necessary to address in order to strengthen the work as a whole.

Response: We greatly appreciate the reviewer's comments on the novelty and significance of our study.

Major comments/concerns:

1. The statistics for many of the behavioral and molecular experiments are not correct. Mann-Whitney U tests should only be used when comparing two groups or variables. However, the authors use it for comparisons of more than two.
 - a. For example, in Figure 1b, f, I, a two-way ANOVA with appropriate post-hoc correction (e.g., Bonferroni) should be used given that there are two variables per group.
 - b. Additionally, for example, in Figure 5a, c, e, and f (but in many other figures as well), a one-way ANOVA with appropriate post-hoc correction (e.g., Tukey or Bonferroni), or a comparable non-parametric analysis such as Kruskal-Wallis test, should be used given that there are 3 groups.

Response: We thank the reviewer for providing such an important suggestion. As suggested, we have performed the statistical analysis for the behavioral and molecular experiments using ANOVA with Tukey post-hoc correction or two-way ANOVA with Bonferroni correction when comparing more than two groups.

2. For the microbiome-related research, it is important to provide more details about the housing conditions. In particular, the authors should state whether the *Chd8*^{+/+} and *Chd8*^{+/-} mice were housed together or separated by genotype after weaning given

recent reports that such conditions may impact both the microbiome and behavior of animals. If the *Chd8*^{+/-} and *Chd8*^{+/+} mice are housed separately, the authors should co-house these mice to determine whether the behavioral deficits are purely microbial driven.

Response: Thanks for this comment. *Chd8*^{+/+} and *Chd8*^{+/-} mice were housed separately by genotype after weaning since housing conditions may impact the gut microbiota and the gut microbiota modulates the behavior of animals. We have added the description of housing conditions in the Materials and Methods under the ‘Mouse maintenance and breeding’ subsection.

Following the reviewer’s suggestion, we co-housed the *Chd8*^{+/+} and *Chd8*^{+/-} mice after weaning in a 3:1 ratio to correct the gut microbiota of *Chd8*^{+/-} mice by *Chd8*^{+/+} mice. After co-housing the mice for four weeks, we checked the behaviors and the gut microbiome of *Chd8*^{+/+} mice, *Chd8*^{+/-} mice, and co-housed *Chd8*^{+/-} mice. We found that co-housing indeed improved the anxiety-like behavior and social interaction behavior but failed to improve the learning and memory deficits of *Chd8*^{+/-} mice (comparison between *Chd8*^{+/-} mice and co-housed *Chd8*^{+/-} mice). We further found that co-housing increased the level of *B. uniformis* in the feces of *Chd8*^{+/-} mice. We focused on the level of *B. uniformis* as the indicator of gut microbiota, given that *B. uniformis* has shown the largest fold change in the gut microbiota of *Chd8*^{+/-} mice and has been proved to improve the ASD-like phenotypes in our study. We have added these data and the relevant description of the results and methods in the revised manuscript (**Fig. 4a-d**).

3. It is not clear whether the increased levels of the amino acid transporters are due to the genetic mutation or the changes in the microbiome. To answer this question, the authors could transfer the microbiome from the WT and mutant mice into germ-free (GF) mice. Alternatively, the authors could make the *Chd8*^{+/-} mice GF and measure levels of amino acid transporters in the GF-*Chd8*^{+/-} mice.

Response: As suggested, we transferred the feces from *Chd8*^{+/+} and *Chd8*^{+/-} mice to the GF mice, respectively, and measured the level of amino acid transporters in the small intestines after fecal microbiota transplantation. We observed no significant alterations of amino acid transporters (*slc6a19* and *slc7a8*) in mRNA level or protein level, suggesting the increased level of intestinal amino acid transporters was not due to dysbiosis of the gut microbiota but due to the genetic mutation. In the revised manuscript, we added these data (**Extended Data Fig. 6h-k**) and the relevant description in the Materials and Methods under the subsection ‘Fecal microbiota transplantation’.

4. ①Are there any changes in amino acid transporters in the brain or ②blood brain barrier permeability in *Chd8*^{+/-} mice? ③Moreover, does *B. uniformis* treatment affect brain amino acid transporter levels? Previously, changes in the levels of amino acid transporters in the blood brain barrier have been shown to lead behavioral symptoms of ASD (Tărlungeanu et al. Cell 2016). Moreover, the authors do not show whether the

inhibitory effect of S-Benzyl-L-cysteine could be working in the brain.

Response:

① We evaluated the levels of 31 amino acid transporters in the adult brain of *Chd8*^{+/-} mice using scRNA-seq. In GABAergic neurons and oligodendrocytes, there were no significant changes in the levels of amino acid transporters. One transporter increased, and one decreased in one subtype of microglia. Seven amino acid transporters showed decreased levels in astrocytes, and one showed increased levels (**Figure R1**) (adjusted $P < 0.05$). Overall, the levels of amino acid transporters in the brain of *Chd8*^{+/-} mice did not show much changes.

Figure R1. Fold changes of amino acid transporters in the adult brain of *Chd8*^{+/-} mice from scRNA-seq.

② Since blood-brain barrier (BBB) permeability is one of the mechanisms of the gut-brain axis (Kim, 2008), we tested BBB permeability and found no significant changes in *Chd8*^{+/-} mice. We added these data (**Extended Data Fig. 4i**) and the relevant description in the Materials and Methods under the subsection ‘Blood-brain barrier (BBB) integrity detection’ in the revised manuscript.

③ Given that changes in the level of large neutral amino acid transporter LAT1 (*Slc7a5*) in the blood-brain barrier have been shown to lead to behavioral symptoms of ASD, we detected whether treatment of *B. uniformis* modulates the expression of *Slc7a5*. We found no significant difference in the expression level of *Slc7a5* in *B. uniformis* gavaged *Chd8*^{+/-} mice. We did not test other amino acid transporters in the brain after *B. uniformis* treatment, as these transporters have not been shown to affect ASD-like phenotypes in *Chd8*^{+/-} mice.

Figure R2. qPCR analysis of *Slc7a5* in the brain.

5. Is *Chd8* expressed in immune cells? Does loss of *Chd8* in immune cells drive inflammation in this model?

Response: Given that *Chd8*^{+/-} mice showed increased levels of inflammatory cytokines in their intestines, we have evaluated the expression of *Chd8* in small intestines at the

adult and embryonic stages from our scRNA-seq data. At the embryonic and adult stage, the expressions of *Chd8* in almost all the immune cells (B cell, T cell, and macrophage at the adult stage and innate lymphoid at the embryonic stage) were extremely low compared with marker genes of each cell type and housekeeping gene *Gapdh* (**Figure R3**), the number in the heatmap indicates the median gene expression). At the embryonic stage, *Chd8* was expressed in several cell types such as enteroendocrine cells but not in immune cells. At the adult stage, *Chd8* showed extremely low expression in almost all the cell types. At this stage, we cannot detect the *Chd8* expression at the protein level (the full-length isoform or smaller isoform) in the small intestine.

Figure R3. The expression of *Chd8*, marker genes, and the housekeeping gene in immune cells at the embryonic (a) and adult stage (b) from scRNA-seq.

6. The authors' claim that increased levels of alpha defensins contribute to gut microbiota dysbiosis in *Chd8*^{+/-} is not supported by the evidence shown. To support this, the authors need to test whether inhibition of alpha defensins in *Chd8*^{+/-} mice normalizes the microbiome.

Response: We thank the reviewer for pointing this out. As suggested, we used an inhibitor of matrix metalloproteinase (MMP), GM6001, which can suppress alpha defensin precursors from being cleaved and activated by MMP7 (Ayabe et al., 2002; Wilson et al., 2009), and thus it inhibits the production of alpha defensins. We found that *Chd8*^{+/-} mice intraperitoneally injected with GM6001 showed increased intestinal alpha defensin levels and decreased *B. uniformis* level. In the revised manuscript, we have added these data (**Fig. 3m-n** and **Extended Data Fig. 4o**) and the relevant description in the Materials and Methods under the subsection 'GM6001 administration'.

7. Did the authors look at behavior of the defensin alpha 1 treated mice? Is increased defensin alpha 1 sufficient to cause behavioral deficits?

Response: We examined the behaviors of defensins-treated mice, but did not observe significant behavioral abnormalities in these alpha defensins gavaged mice (**Figure R4**), although these mice showed a slightly reduced level of *B. uniformis* in their feces (**Fig. 3I**). A possible explanation is that such slight changes on gut microbiota are not sufficient to affect the behavioral phenotypes in wild type mice.

Figure R4. Behavioral tests of alpha defensins gavaged mice. **a**, Percentages of time spent in the light box in the light/dark box test. **b**, Percentages of interaction time (left) and the preference index (right) in the social novelty preference session of the three-chamber social interaction test. Quantitative data are shown as the mean \pm SEM. Statistical analysis was determined by the Mann-Whitney test (**a** and **b** (right panel)) and two-way ANOVA with Turkey's test for multiple comparisons (**b** (left panel)). Significance was indicated by *P* value.

8. Does treatment with *B. uniformis* rescue the changes in the gut morphology, gut microbiota, and inflammatory markers of the *Chd8*^{+/-} mice which the authors describe in figures 1 and 3?

Response: We did not examine the effect of *B. uniformis* treatment on gut morphology, as it does not seem to directly affect neural activities and CNS-related behaviors. As for the gut microbiota, because our study has shown that *B. uniformis* has undergone the largest fold change in the gut microbiota of *Chd8*^{+/-} mice and improves the ASD-like phenotypes, we focused on the level of *B. uniformis* as the indicator of the gut microbiota. We found that treatment with *B. uniformis* could rescue the level of *B. uniformis* in the feces (**Extended Data Fig. 5c**).

As for the inflammatory markers, we found that *B. uniformis* failed to rescue the levels of IL-17A or IL-1 β in small intestines (**Figure R5**). We did not examine other inflammatory markers increased in the *Chd8*^{+/-} mice since we did not focus on the inflammatory effect in this study.

Figure R5. ELISA analysis of cytokine levels in the intestine.

9. The authors claim that the increase in serum L-glutamine drives the social deficits and anxiety-related behaviors in the *Chd8*^{+/-}. However, the data supporting this are correlational. Does treatment with L-glutamine cause social deficits and anxiety-like behavior in WT mice?

Response: We thank the reviewer for this comment. We have tried to increase the serum glutamine level by supplementation with L-glutamine in the drinking water of *Chd8*^{+/+} mice. However, the glutamine level in serum remained stable even though we doubled

the concentration of L-glutamine in the water (**Extended Data Fig. 7b**). We further found that the high concentration of L-glutamine in the drinking water significantly reduced the expression of intestinal amino acid transporters (**Extended Data Fig. 7c**), which may explain why oral supplementation of L-glutamine does not increase serum L-glutamine in *Chd8*^{+/+} mice and thus cannot cause social deficits and anxiety-like behaviors in *Chd8*^{+/+} mice (**Figure R6**). However, genetic mutation of *Chd8* leads to the increased expression of amino acid transporters in the intestine, which may facilitate the transport of glutamine to serum in *Chd8*^{+/-} mice.

Figure R6. Behavioral tests of glutamine-treated mice. **a**, Percentages of time spent in the light box in the light/dark box test. **b**, Percentages of interaction time (left) and the preference index (right) in the social novelty preference session of the three-chamber social interaction test. Quantitative data are shown as the mean \pm SEM. Statistical analysis was determined by the Mann-Whitney test (**a** and **b** (right panel)) and two-way ANOVA with Turkey's test for multiple comparisons (**b** (left panel)). Significance was indicated by *P* value.

10. For the microbiome data, the authors should use PhILR-transformed or CLR-transformed Euclidean distances in addition bray-curtis distances when measuring beta-diversity and performing ordination. Additionally, the authors should consider using ASV outputs instead of the 99% OTUs for looking at taxonomy from the 16S data in order to match the standards methods of the field.

Response: We thank the reviewer for this kind suggestion. We have incorrectly used the 'OTU' description in the previous manuscript. In fact, DADA2 in QIIME2 (version: 2020.2.0) was used to analyze 16s data in our study. The output of DADA2 is an amplicon sequence variation (ASV) table, not an OTU table. For taxonomy annotation, following the recommendations of the QIIME2, we used the blast-based classification method (classify-consensus-blast) to compare the representative sequence to a reference database of sequences with known taxonomic composition (Greengenes 99% OTUs database, version:13_8). We have revised the manuscript accordingly.

In the previous manuscript, we used total sum scaling-transformed bray-curtis distances in the PCA and distance analysis. As suggested, CLR-transformed Euclidean distances were used by *transform(transform = "clr")* in R package *microbiome*, which generated a very similar result (**Figure R7**). We have updated the relevant diagram and description in the revised manuscript (**Extended Data Fig. 4a, k, m and n**).

Figure R7. Comparison between Total Sum Scaling-Transformed and Centered Log-Ratio-Transformed distances.

Minor comments:

- On page 2 paragraph 2, the authors state “A recent study found that the gut microbiota from ASD patients is sufficient to trigger the disease in healthy individuals” citing the work from Sharon et al., Cell 2019. The statement is a mischaracterization of the findings and the overall results/interpretations of the FMT experiments from this paper are highly controversial.

Response: Thanks for the suggestion. We have deleted that citation.

- For the open field test, “time in center” is a preferred measurement of anxiety-like behavior compared to “distance travelled in center.” The authors should consider showing data for “time in center” as well.

Response: As suggested by the reviewer, we have changed the measurement of anxiety-like behavior to “time in center”, and modified the figures and manuscript accordingly.

- On page 4 paragraph 3, the authors say that they examine “region-specific” changes in amino acids by comparing levels in the cerebrum and brain stem. However, this is not what is typically considered region-specific. Instead, examining changes in areas within the cerebrum such as cortex, hippocampus, hypothalamus, etc. is what is considered region specific. Given that the mEPSCs and mIPSCs were recored in cortical neurons, it would make sense to look for region-specific changes in amino acids in the cortex as well.

Response: We thank the reviewer for the comment. Since the cerebrum is not typically considered region-specific, we have changed the expression of “region-specific” to “different parts of the brain” and modified the relevant description in our new manuscript.

- On page 2, paragraph 3, the authors claim that their findings represent “a new mode of communication of the gut-brain axis is ASD” distinct from “neuroactive compounds.”

However, changes in amino acids in the brain due to the microbiome-mediated changes in circulating amino acids have already been shown to underly the anti-epileptic effects of the ketogenic diet/probiotic species (Olson et al., Cell 2018) and changes in the levels of other metabolites such as 5-aminovaleric acid has also been shown to improve some ASD-related behaviors (Sharon et al., Cell 2019). Moreover, changes in brain amino acids due to changes in amino acid transporters has been shown in ASD (Tărlungeanu et al. Cell 2016). While the data the authors show are novel, they do not necessarily constitute an entirely new gut-microbiome-brain pathway as they claim.

Response: We thank the reviewer for pointing this out. As suggested, we tuned down the claim and modified the related description in our revised manuscript.

- On page 3, paragraph 3: the phrase “non-significantly different social abilities” is not very clear/fluent. The authors should consider describing the result as “normal sociability” or “normal preference for social over non-social interaction.”

Response: Thank you for this suggestion. Following the reviewer’s advice, we have changed the phrase “non-significantly different social abilities” to “normal sociability” in our new manuscript.

- On page 3, paragraph 2: While there is a clear trend in intestinal transit time data (Ext Data Fig 1e), the difference between groups is not statistically significant. Therefore, the authors should consider rephrasing “slightly decreased intestinal motility” as “a trend towards decreased intestinal motility” in order to more accurately convey the results.

Response: As suggested by the reviewer, we have rephrased “slightly decreased intestinal motility” as “a trend towards decreased intestinal motility” in our new manuscript.

Reviewer #2 (Remarks to the Author):

Yu et al present a very interesting manuscript entitled *Bacteroides uniformis* restores ASD-like phenotypes by reducing intestinal amino acid transport. In this study, the authors show evidence that CHD8 het mutants display abnormal behaviors, show changes in the gut and in gut flora, and that these microbiota changes in the gut might mechanistically result in the changes in behavioral phenotypes. The study utilizes an array of approaches and reveals several novel findings further supporting a critical role for the gut-brain axis in the regulation of neuro-psychiatric relevant behaviors. Enthusiasm is somewhat lessened by a number of concerns detailed below regarding the proposed mechanisms. Specifically, the mechanism proposed is that AA transporters are altered in the gut, resulting in increased serum glutamine, that results in increased brain glutamine and then glutamate and resulting E/I imbalance. However, in Figures 6c-d, serum glutamine does not correlate necessarily with brain glutamine. Also, as

brain glutamine is significantly, if not mostly, contributed by astrocytic production of glutamine, it will be important to show greater evidence that the proposed mechanism is the driver of the E/I imbalance observed in these animals. Also, glutamine levels in various experiments are quite variable. Overlapping levels are both associated with normal behaviors and with abnormal behaviors, which further raises questions about the proposed mechanism. Also, statistical measures do not seem ideally utilized for the studies performed. The authors should also be cautioned not to define some of the behaviors examined as being equivalent to anxiety or depression: depression- or anxiety-like perhaps would be more appropriate in this context.

Response: We greatly appreciate the reviewer's comments on the novelty and significance of our study. In this revised version, we extensively revised the manuscript and added more experiments and data analyses. Please refer to the following responses for details.

Specific Comments

Figure 1. ①Please include size markers for westerns. ②It is also not clear why the authors check hypothalamus in adult when E14.5 is whole brain? ③Also, What is level of 110kDa isoform at E14.5?

1g: ④For behavior, the authors plot distance in center. What about time in center (which would probably be more comparable with duration in light box of light dark box). ⑤Also, authors should show locomotor activity in OF across time bins (not just total)?

⑥The authors should at least discuss how much of behavior may relate to increased anxiety-like behavior?

⑦The authors show that mutants possess smaller gut (small and large intestine), decreased motility, and decreased weight. How do these factors relate/interact with the subsequent findings re: intestinal flora, absorption, transporters?

Response:

① Thanks for the suggestion. We have included size markers for western blots (**Fig. 1b** and **Extended Data Fig. 1c-d**).

② Since *Chd8* is widely expressed in multiple brain regions, and our *Chd8*^{+/-} mice are not conditional knockout for any specific cell type, we chose the hypothalamus, a brain region that showed relatively high expression for *Chd8* (<http://mouse.brain-map.org/experiment/show/68844247>), to test the expression of *Chd8* in adults. To unify the brain regions, we have selected the cerebral cortex to detect the expression of *Chd8* at different developmental stages (E14.5 and adult) in this revised manuscript. At E14.5, the expression pattern of *Chd8* in the cerebral cortex was similar to that in the whole brain (**Fig. 1b** and **Extended Data Fig. 1c**); in adults, the expression pattern of *Chd8* in the cerebral cortex was similar to that in the hypothalamus (**Fig. 1b** and **Extended Data Fig. 1d**).

③ The 110 kDa isoform did not change significantly in the whole brain or the cerebral cortex at E14.5 (**Extended Data Fig. 1c**).

- ④ As suggested by the reviewers, we have changed the measurement of anxiety-like behavior to “time in center”, and modified the description in our revised manuscript. The “time in center” showed the same trend of changes compared to the “distance travelled in center” (Fig. 1g, Fig. 4g and Fig. 6g).
- ⑤ We have added the locomotor activities in open-field across time bins (Extended Data Fig. 1l and r) in our revised manuscript.
- ⑥ The social deficits in *Chd8*^{+/-} mice may be related to anxiety-like behavior. Epidemiological surveys and animal models show a high rate of co-morbidity between ASD and anxiety-like behaviors. One possible reason that the social interaction behavior and the anxiety-like behavior are related is that they are modulated by the same neural circuit (Allsop et al., 2014). In addition, cognitive disturbances in *Chd8*^{+/-} mice can be associated with their anxiety-like behavior. It has been shown that anxiety may compromise performance on cognition-based tasks (Maloney et al., 2014).
- ⑦ Some phenotypes, especially the phenotypes in the intestines, may be related to the following findings. Since the change in gastrointestinal transit is supposed to affect the delivery rate of nutrients to the gut microbiota (Rhee et al., 2009), the host may regulate the gut microbiota indirectly by changing the luminal environment in *Chd8*^{+/-} mice. Both the length of the intestine and the intestinal transit of diets may impact the absorption of the nutrients by altering the absorption area and available time for nutrients to contact intestinal epithelia.

Figure 2: In figure 2 g,h: ①how is quantification of expression performed? From plots, it appears that more *slc*__ expressing cells are present. It’s not clear if the intensity of the color scheme denotes level of expression or not, but it’s not clear that level of expression vs. number of cells expressing the transporters has been altered. ②In immuno’s in 2i, the figure suggests a massive increase in transporter levels (almost all or nothing, especially for 6A19). However, in the quantification, it is merely < 1.5fold increase. The figure does not appear to be representative.

Re: ③metabolic studies: to my understanding, glutamine is mostly produced from muscle for use in the body while most of brain glutamine is locally synthesized in astrocytes. It will be important to show that in fact brain levels are directly impacted by these gut transporters (the gavage experiments and the transporter blocker experiments are consistent with the hypothesis but aren’t wholly convincing that alternative mechanisms aren’t at play. ④Also, *slc6a19* transports a number of AA’s including glutamine, leucine, isoleucine, meth. However, only glut is increased. Why do the authors feel that a change in the transporter would result in a specific increase? Also, for *Slc7a8*, alanine, serine, threonine, cysteine, phenylalanine, tyrosine, leucine, arginine and tryptophan should be transported. Again, of these, the only AA that is increased is tryptophan. This mechanism should either be discussed at minimum or experimentally evaluated.

Response:

① In Fig2.g,h, the density of red color indicates the expression level of *Slc6a19* and *Slc7a8* based on log-transformed normalized UMI counts. The barplot shows the

differences in gene expression abundance between the two groups based on the Wilcoxon test and adjusted by the BH method. According to your suggestion, we added the analysis of cell number by `chisq.test` in R. The results showed that expression levels and numbers of cells expressing the transporters have been increased in *Chd8*^{+/-} mice (**Extended Data Fig. 3**).

② We re-performed the Immunofluorescence experiments and used representative figures (**Fig. 2i**) in our revised manuscript.

③ Thanks for the valuable comment. Although glutamine can be synthesized de novo in almost all tissues in the body, its dietary intake contributes to a substantial proportion, but not all, of glutamine levels in the serum (Labow and Souba, 2000; Ducroc et al., 2010). As mentioned by the reviewer, glutamate released into the synaptic cleft is transported into astrocytes, where glutamate is converted into glutamine, and glutamine is then shuttled to neurons for glutamate production. This process has been well known as the glutamate-glutamine cycle. However, glutamine can also be imported at the blood-brain barrier through amino acid transporters (Xiang et al., 2003). Our data (the *B. uniformis* gavage experiment and the intestinal transporters inhibitor treatment) provide some evidence that the intestinal transporters can modulate the brain glutamine level. However, we cannot rule out alternative mechanisms, such as via vagus nerve and/or neural active microbial metabolites. We have added the related discussion in the Discussion section.

④ We thank the reviewer for this important comment. We agree with the reviewer that amino acids and their transporters are not in a strict one-to-one relationship. One possible reason that several, but not all, neutral amino acids changed their serum levels in *Chd8*^{+/-} mice may be due to the fact that not all the neutral amino acid transporters were increased in the intestine. For neutral amino acid transporters including *Slc6a19*, *Slc7a8*, *Slc7a15*, *Slc7a7*, *Slc1a5*, *Slc3a2*, *Slc6a14*, *Slc7a6* and *Slc38a2*, six of them maintained unchanged expression, while three of them (*Slc6a19*, *Slc7a8* and *Slc7a15*) showed increased expression levels (**Extended Data Fig. 3b**). Another possible reason may be related to the homeostasis of serum amino acid levels. Besides intestinal amino acid transporters, some other pathways, including intra-organ exchange of amino acids (Liao et al., 2018) and transport of amino acids from the kidney, can affect serum amino acid levels (Broer, 2008). In addition, the transporter preference for some specific amino acids (Schioth et al., 2013), the concentration of substrates in intestinal contents, may also contribute to the uptake of amino acids in intestines. We have added the related discussion in the Discussion section.

Fig3. The *Chd8*^{+/-} mice showed a decrease in the abundance of *Bacteroides* in the gut microbiota, but this decrease was not found in the defensin α 1-treated mice (Extended Data Fig. 4j). If this is the proposed mechanism by which *Bacteroides* is decreased in mutant mice, shouldn't a decrease be reproduced? Also when you infuse defensin, *Firmicutes* goes up (this is not a species examined/observed in 3b)? In essence, the change in bacterial flora (mutant vs. defensin) have some overlap but are quite distinct.

Response: The decreased abundance of *Bacteroides* has not been replicated in defensin

α 1-treated mice even if changes in the microbiota have been observed, implying the defensin α 1 alone may not be sufficient to induce the ASD-like microbiota. Since both defensin α 1 and α 2 were elevated in mutant mice, we further treated the *Chd8*^{+/+} mice with the combination of defensin α 1 and α 2 by gavage and found that these mice showed a decreased level of *B. uniformis* in their feces ($P < 0.05$). These results indicate that the influence of defensins on the microbiota, especially the level of *B. uniformis*, is a cumulative effect of multiple defensins, and it is difficult for a single defensin treatment to trigger the ASD-like changes in the microbiota.

The reasons that *Firmicutes* goes up in defensin α 1-treated mice but not in *Chd8*^{+/-} mice may be as follows. Multiple contributors may affect the microbiota in *Chd8*^{+/-} mice, such as defensins, inflammatory cytokines, immunoglobulin A, and innate lymphoid cells. Other factors may balance the impact of defensin α 1 on the *Firmicutes*, and thus *Chd8*^{+/-} mice did not show increased *Firmicutes* in their feces.

Fig4. ① With *B. uniformis* gavage, I may have missed this, but did this procedure actually increase *B. uniformis* abundance, especially relative to other species? Does this manipulation impact the overall gut flora? Also, is the observed effect specific to *B. uniformis*? If you infuse another species, firmicutes for instance which goes up when you put in defensin, do you observe a similar impact?

② 4g: Why are ratios of metabolites so different from figure 2 (glu/gaba and levels of glutamine)? Mutant levels are lower here than in controls in fig2. If levels/abundance are the critical measure to generate E/I imbalance, why is this lower level still associated with E/I imbalance and why is it not imbalanced in controls in 2 when the glutamine levels are much higher?

③ 4f: mIPSC appears unchanged. Do authors propose that increase in mEPSC freq is the driver of the E/I phenotype?

④ For this figure, it will be important to show locomotor behaviors. ⑤ The authors state that their findings show “increased anxiety” 4c-d. I think the data might argue the opposite. ⑥ Authors also use the word restore which may not be the proper term when improving behavior. Authors should also be cautioned about these behaviors equating to anxiety or depression. They are at best a test for anxiety-like behaviors.

Response:

① We have examined the level of *B. uniformis* by qPCR after the treatment of *B. uniformis*. As shown by our results, the gavage of *B. uniformis* increased the level of *B. uniformis* in the feces of *Chd8*^{+/-} mice. We did not examine the changes of other species since only *Bacteroides* changes with *B. uniformis* showing the largest fold change in the feces of *Chd8*^{+/-} mice. We speculate that some other species in genera *Bacteroides* may have similar effect on the ASD-relevant phenotypes, as they exhibited remarkably reduced abundance in the *Chd8*^{+/-} mice.

② Indeed, the level of metabolites in Fig. 4 cannot be directly compared to that in Fig. 2 due to batch effect. Considering the differences on sample preparation, specimen processing, preservation time, and standard curves used in targeted metabolomics, some discrepancies are reasonable between different batches of metabolomic assays.

- ③ We can see from Fig.4f (Fig. 4j in revised manuscript) that neither the frequency nor the amplitude of mIPSC was changed between *Chd8*^{+/-} mice gavaged with *B. uniformis*, and *Chd8*^{+/-} mice gavaged with PBS, but the frequency of mIPSC was reduced in *Chd8*^{+/-} mice gavaged with PBS compared with *Chd8*^{+/+} mice gavaged with PBS. According to our data, an increased mEPSC frequency (**Fig. 2a** and **Fig. 4i**) and a decreased mIPSC frequency (**Fig. 2b** and **Fig. 4j**) were observed in *Chd8*^{+/-} mice compared with *Chd8*^{+/+} mice, implying the E/I imbalance in *Chd8*^{+/-} mice may result from abnormality in both mEPSC and mIPSC. However, gavage with *B. uniformis* only rescued the frequency of mEPSC of *Chd8*^{+/-} mice, but not mIPSC.
- ④ We have added locomotor behaviors and the related description in our revised manuscript (**Extended Data Fig. 5f**).
- ⑤ Sorry for the mistake. We have corrected the expression of “increased anxiety” to “decreased anxiety-like behavior” (**Fig. 4g-h**).
- ⑥ We thank the reviewer for providing such an important suggestion. Following the reviewer’s suggestion, we have modified the description of relative behaviors and replaced the “anxiety behavior” with “anxiety-like behavior”.

Fig5: ① again, like in figure 4, in mutant, serum levels of glutamine here are comparable to those in controls in Figure 2. Yet, here they are abnormal, but there, normal.

② 5e: Like with Figure 2, fluorescence intensity here for slc6a19 is quantified as less than a 2 fold increase, yet staining image is from 0 to extremely bright. Either this is not a fair representative increase or the graphs are not representative. In the individual values, the greatest increase is 3 fold. Again, does not match image. Similarly, in f, western for slc6a19 increase is higher in bar graph but appears to be less increased than increased for slc7a8 in blot shown)

③ Also, as with defensin gavage studies, AA changed with *B. uniformis* gavage are all very different than changed in mutant. Glutamine is changed, although it’s not clear that it’s significantly changed from control. Also, tryptophan, which was the other AA changed in mutant, is not increased while others appear to be changed that weren’t changed in mutant.

④ This is relevant to the entire study, the authors use a Mann-whitney test here. It is most likely not the most appropriate test to use. This test should be used for comparison of 2 populations. For all comparisons here, there are three populations to be compared. This applies to many of the figures and to all social interaction figures which should not only have comparison between novel animal and familiar animal, but also, how that differs from control.

Response:

- ① These experiments were performed in different batches. Considering the differences on sample preparation, specimen processing, preservation time, and standard curves used in targeted metabolomics, some discrepancies are reasonable between different batches of metabolomic assays.
- ② We re-performed the Immunofluorescence and Western blot experiments and used

representative figures in **Fig.5 e-f**.

③ The amino acids changed with *B. uniformis* gavage are different from those changed in *Chd8*^{+/-} mice, both in the serum and the feces. We speculate that this may be due to the relatively small sample size along with the large within-group variations, especially for the group of *Chd8*^{+/+} mice gavaged with PBS. Therefore, in this revised manuscript, we increased the sample size, and found that serum glutamine and tryptophane levels increased in PBS-gavaged *Chd8*^{+/-} mice compared with PBS-gavaged *Chd8*^{+/+} mice, decreased in *B. uniformis*-gavaged *Chd8*^{+/-} mice compared with PBS-gavaged *Chd8*^{+/-} mice, and did not change in *B. uniformis*-gavaged *Chd8*^{+/-} mice compared with PBS-gavaged *Chd8*^{+/+} mice (**Fig. 5a** and **Extended Data Fig. 6a**). Similarly, we compared the differences within the three groups (*Chd8*^{+/+} mice gavage with PBS, *Chd8*^{+/-} mice gavage with PBS, *Chd8*^{+/-} mice gavage with *B. uniformis*) for all the other amino acids in the serum (**Extended Data Fig. 6a**) and the feces (**Extended Data Fig. 6g**).

④ We have re-performed the statistical analysis in all figures that involved more than two groups with the appropriate tests suggested by the reviewer and updated the *P* values. Also, the comparison with the control group in all figures involving more than three groups was included in our revised manuscript.

Figure 6: ① What is function of s benzyl L cysteine? Authors do not show proof that it inhibits AA transporter, only that it reduced serum glutamine. ② Also, does it impact anything else or is its actions specific to the transporter? ③ Also, administration does not decrease brain levels despite decrease in serum. So if functions at the intestine only, then why does it impact mutant but not control (i vs. d). and if the proposed mechanism that brain follows serum, it's not clear why this disconnect is present. ④ Also, why do authors evaluate cerebrum in one place and brain in another? ⑤ Here, glutamine levels in controls are higher than glutamine levels of mutant in fig2 and 4. Again, why do these animals not have E/I imbalance or abnormalities? ⑥ Also, what is E/I imbalance/electrophysiology in this paradigm? ⑦ Again, stats need to be performed correctly.

Response:

① S-Benzyl-L-cysteine is an inhibitor of amino acid transporter SLC6A19 and serves as the substrate analog to SLC6A19 (Cheng et al., 2017). It has been previously reported that S-Benzyl-L-cysteine can inhibit SLC6A19 at the cellular level (Cheng et al., 2017). Considering it is difficult to examine the inhibitory effect of S-Benzyl-L-cysteine on SLC6A19 *in vivo*, we only measured the serum glutamine level as an indicator.

② By screening the inhibitory effect of several commercially available substrate analogs on SLC6A19, Cheng et al. found that S-Benzyl-L-cysteine showed the highest inhibitory effect on SLC6A19 (Cheng et al., 2017). In addition, S-Benzyl-L-cysteine showed an inhibition effect on several other amino acid transporters (e.g., sodium non-dependent transporters) (Cheng et al., 2017). Another commercially available substrate analog, benztropine, had been found to be more specific for SLC6A19 than S-Benzyl-L-cysteine. However, benztropine is an anticholinergic drug (Gelenberg et al., 1989) and may have some side effects on the central nervous systems. Therefore, we chose S-

Benzyl-L-cysteine instead of benztropine in our experiments.

③ The wild type mice supplemented with S-Benzyl-L-cysteine showed a slightly reduced glutamine level in the brain ($p=0.063$). We think the duration of S-Benzyl-L-cysteine administration may play an important role in the effect of the drug. The treatment time with S-Benzyl-L-cysteine on *Chd8*^{+/-} mice (four weeks) was longer than on *Chd8*^{+/+} mice (two weeks). Since we aimed to test the effect of S-Benzyl-L-cysteine on the serum glutamine level, we did not explore the long-term effect of S-Benzyl-L-cysteine treatment (e.g. four weeks) on brain glutamine level in wild type mice.

④ We found that the changes of amino acids in the cerebrum of *Chd8*^{+/-} mice are similar to those in the whole brain (**Fig. 2c** v.s. **Extended Data Fig. 2b**; **Fig. 4k** v.s. **Extended Data Fig. 5g**). Also, since we focused on the level of amino acids in *Chd8*^{+/-} mice, we used its level in the whole brain to estimate the effect of S-Benzyl-L-cysteine on *Chd8*^{+/+} mice. Similarly, the cerebrum was used when evaluating the level of amino acids in the brain of *Chd8*^{+/-} mice.

⑤ Indeed, the level of metabolites in Fig. 6 cannot be directly compared to that in Fig. 2 and Fig. 4 due to batch effect. Considering the differences on sample preparation, preservation time, and standard curves used in targeted metabolomics, some discrepancies are reasonable when comparing different batches of metabolomic assays.

⑥ We did not detect the E/I imbalance at the electrophysiological level. As shown in **Fig. 4i** and **Fig. 4j**, *B. uniformis* only partially decreased the frequency of mEPSC and had no effect on mIPSC. Since the relatively weak impact of *B. uniformis* on electrophysiology and similar mechanisms were assumed between *B. uniformis* and S-Benzyl-L-cysteine, we only performed behavioral tests, but did not perform mEPSC or mIPSC tests on *Chd8*^{+/-} mice treated with S-Benzyl-L-cysteine.

⑦ We have re-performed the statistical analyses in all figures that involved more than two groups as suggested by the reviewer and modified the *P* value and relative description in our new manuscript.

Reviewer #3 (Remarks to the Author):

In this work, Yu et al investigate the effect of the chromatin remodeler Chd8 on interactions between the microbiome and the host, focusing on effects related to ASD and gastrointestinal functions. The authors perform an in-depth study, establishing effects on a wide number of behavioural, neurological and physiological effects, as well as effects on microbiome composition and function.

I am quite excited about this study! The study is both novel and significant for the field, as it establishes mechanistic relationships between host genotype, the microbiome and host phenotypes. This is exactly the kind of detailed work that the microbiome field needs, as there currently is a large amount of associative data, but we are only scraping the surface in terms of cause-effects relations in terms of the effect that the microbiome has on host physiology.

The study is very thorough, and mostly supports its claims through multiple assays and

rigorous analysis using appropriate methods. The data analysis is thorough and sound. The manuscript is overall very well-written.

Response: We greatly appreciate the reviewer's comments on the novelty and significance of our study.

Comments:

1. The main issue I have with the study is the claims regarding defensin-1. The data presented does not support that changes in microbiome composition require defensin-1 and are not significantly different. It is also unclear why not all conditions are analysed in parallel and shown in the same panel. I suggest the data can be shown but that the claim is removed – overall the study is so informative that the overall conclusions will not be affected.

Response: We thank the reviewer for this valuable suggestion. The decreased abundance of *Bacteroides* has not been replicated in defensin $\alpha 1$ -treated mice, although changes in the microbiota have been observed, implicating the defensin $\alpha 1$ alone may not be sufficient to induce the ASD-like microbiota. Since both defensin $\alpha 1$ and $\alpha 2$ were elevated, we further treated the *Chd8*^{+/+} mice with defensin $\alpha 1$ and $\alpha 2$ by gavage and found that these mice showed a slightly decreased level of *B. uniformis* in their feces (**Fig. 3I**). These results indicate that the influence of defensins on the microbiota, especially the level of *B. uniformis*, may be a cumulative effect of multiple defensins, and thus it is difficult for a single defensin treatment to trigger the ASD-like changes in the microbiota.

2. The authors do not seem to have measured if feeding behaviour is affected by in *Chd8*^{+/-}. A change in feeding could explain many if not all of the phenotypes so needs to be measured.

Response: As suggested, we evaluated the feeding behavior of mice by calculating the food consumption normalized by body weight for each cage of mice (Ghule et al., 2020), and found that *Chd8*^{+/-} mice did not show significant changes in food intake. In the revised manuscript, we added these data (**Extended Data Fig. 1g**) and the relevant description in the Materials and Methods under the subsection 'Assessment of feeding behavior'.

3. p. 3: 'The *Chd8*^{+/-} mice exhibited non-significantly different social abilities': I found this sentence confusing – can it be made clearer what the authors mean? Text needs to be clearer to explain that there are behavioural differences, as written now it sounds like they observe differences that are not significant.

Response: Thank you for this suggestion. We have changed the sentence "The *Chd8*^{+/-} mice exhibited non-significantly different social abilities" to "The *Chd8*^{+/-} mice exhibited normal sociability" in our new manuscript.

4. p. 7 ‘These results indicated that the increased levels of defensins and inflammatory factors in the intestine may jointly lead to dysbiosis of the gut microbiota in the *Chd8*^{+/-} mice.’: The authors claim changes in cytokines affect microbial composition, when it could be the other way around. Please change the claim.

Response: The *Chd8*^{+/-} mice showed increased levels of intestinal inflammatory cytokines, suggesting the altered microbial composition in *Chd8*^{+/-} mice may be related to intestinal inflammation since intestinal inflammation can cause gut microbiota dysbiosis (Kamada et al., 2013; Amoroso et al., 2020). Also, since dysbiosis in the gut microbiota can also lead to inflammation (Amoroso et al., 2020), gut microbiota dysbiosis and increased inflammation in *Chd8*^{+/-} mice may reinforce each other. We have modified the relative expression in our new manuscript.

Reference

- Allsop, S.A., Vander Weele, C.M., Wichmann, R., and Tye, K.M. (2014). Optogenetic insights on the relationship between anxiety-related behaviors and social deficits. *Front Behav Neurosci* 8, 241.
- Amoroso, C., Perillo, F., Strati, F., Fantini, M.C., Caprioli, F., and Facciotti, F. (2020). The Role of Gut Microbiota Biomodulators on Mucosal Immunity and Intestinal Inflammation. *Cells* 9.
- Ayabe, T., Satchell, D.P., Pesendorfer, P., Tanabe, H., Wilson, C.L., Hagen, S.J., and Ouellette, A.J. (2002). Activation of Paneth cell alpha-defensins in mouse small intestine. *J Biol Chem* 277, 5219-5228.
- Broer, S. (2008). Amino acid transport across mammalian intestinal and renal epithelia. *Physiol Rev* 88, 249-286.
- Cheng, Q., Shah, N., Broer, A., Fairweather, S., Jiang, Y., Schmoll, D., Corry, B., and Broer, S. (2017). Identification of novel inhibitors of the amino acid transporter B(0) AT1 (SLC6A19), a potential target to induce protein restriction and to treat type 2 diabetes. *Br J Pharmacol* 174, 468-482.
- Ducroc, R., Sakar, Y., Fanjul, C., Barber, A., Bado, A., and Lostao, M.P. (2010). Luminal leptin inhibits L-glutamine transport in rat small intestine: involvement of ASCT2 and B0AT1. *Am J Physiol Gastrointest Liver Physiol* 299, G179-185.
- Gelenberg, A.J., Van Putten, T., Lavori, P.W., Wojcik, J.D., Falk, W.E., Marder, S., Galvin-Nadeau, M., Spring, B., Mohs, R.C., and Brotman, A.W. (1989). Anticholinergic effects on memory: benztropine versus amantadine. *J Clin Psychopharmacol* 9, 180-185.
- Ghule, A., Racz, I., Bilkei-Gorzo, A., Leidmaa, E., Sieburg, M., and Zimmer, A. (2020). Modulation of feeding behavior and metabolism by dynorphin. *Sci Rep* 10, 3821.
- Kamada, N., Seo, S.U., Chen, G.Y., and Nunez, G. (2013). Role of the gut microbiota in immunity and inflammatory disease. *Nat Rev Immunol* 13, 321-335.
- Kim, K.S. (2008). Mechanisms of microbial traversal of the blood-brain barrier. *Nat Rev Microbiol* 6, 625-634.
- Labow, B.I., and Souba, W.W. (2000). Glutamine. *World J Surg* 24, 1503-1513.
- Liao, S.F., Regmi, N., and Wu, G. (2018). Homeostatic regulation of plasma amino acid concentrations. *Front Biosci (Landmark Ed)* 23, 640-655.
- Maloney, E.A., Sattizahn, J.R., and Beilock, S.L. (2014). Anxiety and cognition. *Wiley Interdiscip Rev Cogn Sci* 5, 403-411.

- Rhee, S.H., Pothoulakis, C., and Mayer, E.A. (2009). Principles and clinical implications of the brain-gut-enteric microbiota axis. *Nat Rev Gastroenterol Hepatol* 6, 306-314.
- Schioth, H.B., Roshanbin, S., Hagglund, M.G., and Fredriksson, R. (2013). Evolutionary origin of amino acid transporter families SLC32, SLC36 and SLC38 and physiological, pathological and therapeutic aspects. *Mol Aspects Med* 34, 571-585.
- Wilson, C.L., Schmidt, A.P., Pirila, E., Valore, E.V., Ferri, N., Sorsa, T., Ganz, T., and Parks, W.C. (2009). Differential Processing of α - and β -Defensin Precursors by Matrix Metalloproteinase-7 (MMP-7). *J Biol Chem* 284, 8301-8311.
- Xiang, J., Ennis, S.R., Abdelkarim, G.E., Fujisawa, M., Kawai, N., and Keep, R.F. (2003). Glutamine transport at the blood-brain and blood-cerebrospinal fluid barriers. *Neurochem Int* 43, 279-288.

Reviewers' Comments:

Reviewer #1:

Remarks to the Author:

I am very impressed by the effort the authors have taken to reply to all of our comments. This paper is an important contribution to the movement in neuroscience for the transformative idea that complex behaviors, can be specifically affected/modulated by genetic and microbial interactions.

Reviewer #3:

Remarks to the Author:

The authors have added data and revised the manuscript, and addressed my comments.